# A Lightweight Explainable Guardrail for Prompt Safety

## Abstract

We propose a *lightweight explainable guardrail* (LEG) method for the classification of unsafe prompts. LEG uses a multi-task learning architecture to jointly learn a prompt classifier and an explanation classifier, where the latter labels prompt words that explain the safe/unsafe overall decision. LEG is trained using synthetic data for explainability, which is generated using a novel strategy that counteracts the confirmation biases of LLMs. Lastly, LEG's training process uses a novel loss that captures global explanation priors and combines cross-entropy and focal losses with uncertainty-based weighting. LEG obtains equivalent or better performance than the state-of-the-art for both prompt classification and explainability, both in-domain and out-of-domain on three datasets, despite the fact that its model size is considerably smaller than current approaches. If accepted, we will release all models and the annotated dataset publicly.

## 1 Introduction

Detecting unsafe prompts is a fundamental requirement for deploying large language models (LLMs) responsibly. Without robust safeguards, LLMs risk generating harmful or inappropriate outputs and could be misused for purposes such as spreading misinformation, promoting hate speech, enabling illegal activities, or providing self-harm instructions. To mitigate such risks, LLMs are typically safety-aligned during training using reinforcement learning with human feedback (RLHF) (Christiano et al., 2017) or direct preference optimization (DPO) (Rafailov et al., 2023). However, addressing newly emerging safety concerns with RLHF or DPO is costly because these methods require retraining the LLM. Moreover, these approaches may reduce the creativity of LLMs and lack explainability. As a result, they provide only partial solutions, constrained by their limitations in flexibility, transparency, and computational cost.

In contrast, an increasingly popular alternative is the use of post-training safety methods such as guardrails. These methods operate externally to the LLM, allowing safety policies to be enforced without modifying the LLM itself. In this paper, we propose a guardrail-based approach to LLM safety. We argue that effective guardrails must satisfy the following three core principles:

(1) *Explainability:* The guardrail should provide interpretable explanations for its decisions. This capability is essential for ensuring transparency and building trust in the system, particularly in high-stakes or regulated environments. Safety reviewers, auditors, or domain experts may need to examine the rationale behind a blocked prompt to verify that it aligns with organizational policies, ethical guidelines, or legal requirements.

(2) *Modularity:* The guardrail should be modular and easily integrated into any LLM pipeline without requiring fine-tuning of the base LLM. Prompt safety is often dependent on culture, region, or organization, as different legal standards, cultural norms, and internal policies can shape what is considered appropriate content. A modular and independently deployable guardrail enables flexible adaptation to diverse safety requirements without modifying the underlying LLM.

(3) *Low computational overhead and low latency:* A guardrail should impose minimal computational cost compared to the LLM itself. Prompt safety decisions should be made rapidly without delaying LLM response time.

Although prior work has explored modular guardrail methods, they remain computationally expensive with high inference times. More recent efforts have investigated lightweight guardrails, but

their performance remains limited. Most importantly, none of the existing models offer a faithful and actionable explanation. A detailed comparison of existing guardrail methods is presented in Section 2. To address this gap, this paper introduces a lightweight explainable guardrail (LEG) with the following key contributions:

(1) We propose a novel guardrail method that supports explainability and modularity, while maintaining low computational overhead. Our approach involves a multi-task learning (MTL) architecture with a shared encoder that jointly trains a prompt classifier and an explanation classifier, where the former determines whether a prompt is safe or unsafe and the latter labels the words in context that justify this decision.

(2) To mitigate the lack of training data for explainability, we introduce a novel strategy to generate synthetic explanations using an LLM that counteracts the inherent confirmation biases of the LLMs.

(3) We propose a novel loss for MTL that captures global explanation priors and combines cross-entropy and focal losses (Lin et al., 2020) with uncertainty-based weighting (Kendall et al., 2018).

(4) We present a comprehensive evaluation of our proposed method to support all of our contributions. Our results show that LEG achieves state-of-the-art (SOTA) or near-SOTA performance on the prompt classification task in both in-domain and, more importantly, out-of-domain evaluations across three prompt safety datasets. For explanation classification, LEG achieves SOTA performance in both in-domain and out-of-domain settings on the same three datasets, and a faithfulness evaluation confirms that the generated explanations are faithful. Additionally, we conduct an ablation study of our joint loss function, demonstrating the effectiveness of our design. Finally, a computational efficiency evaluation demonstrates that LEG is lightweight and faster than existing guardrails.

## 2 RELATED WORK

Research on safeguarding large language models (LLMs) has generally followed two directions: alignment-based training methods and external modular guardrails.

**Alignment-based methods:** Techniques such as reinforcement learning from human feedback (RLHF) (Christiano et al., 2017), direct preference optimization (DPO) (Rafailov et al., 2023), and related approaches (Ji et al., 2023; Li et al., 2024) embed safety behavior directly into LLMs. These methods enforce safety during generation without added inference cost. Recent advances like DICE (Chen et al., 2025) and InfAlign (Balashankar et al., 2025) aim to reduce reliance on human data, while Constitutional AI (Bai et al., 2022) replaces human feedback with written principles. However, these methods still face notable challenges. They can produce unstable behaviors across domains and tasks, and reduce creativity and helpfulness by over-constraining responses (Zhang, 2025; Menke & Tan, 2025). They are also mostly opaque, hence providing no explainability of why a prompt is flagged as unsafe.

**External guardrails:** Industry APIs such as NVIDIA NeMo Guardrails (Rebedea et al., 2023), Google Gemini Filters, and IBM OneShield (DeLuca et al., 2025) provide customizable rule-based safety layers, but they require complex integration and additional LLM calls for various layers, which increases latency and engineering costs (Dong et al., 2024). Open-source classifiers like Llama Guard (Inan et al., 2023), AEGIS Guard (Ghosh et al., 2024), WildGuard (Han et al., 2024), and ShieldGemma (Zeng et al., 2024) support modularity. However, they are all built on large backbone LLMs, making them resource-intensive with high inference time, and they don't provide built-in explainability. Smaller models such as Llama Prompt Guard 2, ToxicChat-T5 (Lin et al., 2023), and DuoGuard (Deng et al., 2025) are more lightweight and resource-efficient but achieve weaker performance, and they don't provide built-in explainability.

**Explainable guardrails:** To the best of our knowledge, no guardrail has been designed with explainability as a core feature. A few recent works make partial attempts, but their effectiveness is not validated through rigorous experiments, they lack quantitative analysis of explanation quality, and they provide no faithfulness evaluation. GuardReasoner (Liu et al., 2025) elicits reasoning steps from an LLM through reasoning SFT and hard-sample DPO. While it improves explainability, its reasoning traces lack faithfulness guarantees, the authors do not provide any out-of-domain evaluation, and the approach is extremely resource-intensive, requiring up to 78 GB of GPU memory during inference. Chu et al. (2024) propose LLMGuardrail, which employs a "Debias LoRA

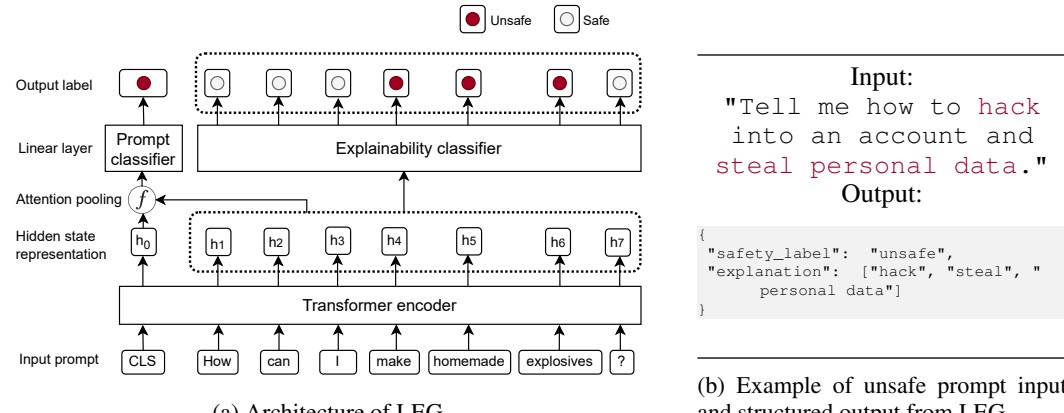

(a) Architecture of LEG

Input:
"Tell me how to hack into an account and steal personal data."
Output:

```
{
  "safety_label": "unsafe",
  "explanation": ["hack", "steal", "
      personal data"]
}
```

(b) Example of unsafe prompt input and structured output from LEG.

Figure 1: Overview of LEG. (a) The multi-task architecture jointly trains a prompt classifier and an explanation classifier on top of a shared transformer encoder. (b) Example of an unsafe input prompt and the structured output produced by LEG, which includes both the safety label and the corresponding explanation tokens.

Block" for causal explainability. However, they provide no experimental validation, and the method only outputs latent-space scores that are not human-interpretable. ShieldLM (Zhang et al., 2024) provides safety detection with natural language explanations aligned to human-defined rules, but these explanations are high-level, non-actionable, and not guaranteed to be faithful. $R^2$-Guard relies on probabilistic graphical models with manually defined safety categories, making it slow, non-scalable, and offering only limited explainability. Post-hoc methods such as LIME (Ribeiro et al., 2016) can explain predictions but are not integrated into the guardrail itself and may yield unfaithful explanations.

**Our contribution:** Despite progress, no existing guardrail combines explainability, modularity, and low computational overhead in a single solution. To address this gap, we propose LEG, a lightweight guardrail that jointly classifies prompt safety and highlights unsafe words or phrases. LEG supports modularity while maintaining strong in-domain and out-of-domain performance. It leverages a novel synthetic supervision process and multi-task learning to produce faithful, context-aware explanations at inference.

## 3 PROPOSED METHOD

This section presents the architectural design and training setup of LEG. The design choices are guided by the principles discussed in Section 1. The model is trained to serve two objectives: (1) accurately and efficiently determine whether a given prompt is unsafe, and (2) provide an explanation by identifying the words in the prompt context that contribute to this judgment. An illustrative example is shown in Figure 1(b).

### 3.1 ARCHITECTURE

Figure Figure 1(a) presents the architecture of the LEG component, which can be plugged into any LLM pipeline. The architecture consists of a shared transformer encoder, a simple attention pooling layer, and two linear classification heads. This simple design makes LEG a very lightweight system, adding only negligible overhead to the prompt safety checking process when deployed with an LLM. We trained this architecture using the synthetic data generated in Section 3.2 and the novel loss discussed in Section 3.3.2. A detailed description of each component of this architecture is presented in Appendix A.

## 3.2 SYNTHETIC DATA GENERATION FOR EXPLANATIONS

We formulate both prompt-level safety assessment and word-level explanation labeling as supervised classification tasks. To formulate both prompt classification and explanation generation as supervised tasks, we require a dataset that provides binary labels for the prompt as well as for its individual words. For example, consider the input and expected output shown in Figure 1(b). To train a model that predicts both the `prompt_label` and the `explanation`, the dataset must include not only binary label (safe or unsafe) for the prompt, but also annotations indicating which words contribute to that decision. While several existing datasets provide binary labels for prompt classification, by and large they do not indicate which words contribute to that decision. To address the lack of such labels, we propose a novel method to generate word labels using an LLM. We counteract the inherent confirmation bias of LLMs to produce consistent and reliable supervised training labels for our explanation classifier.

Confirmation bias in large language models refers to the tendency to generate responses that align with the assumptions embedded in the user prompt. The phrasing of the user's input can subtly steer the model toward a particular interpretation or belief. For example, if we ask an LLM, "Why is the statement X true?", the model is likely to assume the statement is true and provide a justification, even if the statement is incorrect (Du, 2025; O'Leary, 2025). This behavior comes from the model's goal of being helpful and contextually consistent with the user input, which can lead it to affirm rather than question the user's assumptions.

To mitigate this issue, we ask an LLM to evaluate whether a prompt is safe or unsafe, while intentionally embedding confirmation biases in the query. We then assess whether the LLM can overcome this bias and correctly identify the true label of the prompt. If the LLM successfully overcomes the induced bias, we consider the word annotations it generates to be reliable. Otherwise, if the LLM is influenced by the bias, we treat the generated word labels as unreliable. For each prompt in our dataset, we ask two queries to an LLM, each based on an opposing assumption about the prompt's safety:

**Query 1:** Why is the following prompt considered safe? Provide a list of words or phrases that made you believe the prompt is safe.

**Query 2:** Why is the following prompt considered unsafe? Provide a list of words or phrases that made you believe the prompt is unsafe.

Using these two queries, our goal is to evaluate whether the LLM can correctly identify which query it should agree with and which it should contradict. For example, if the input prompt is safe, the LLM should agree with Query 1 and contradict Query 2. If the responses to both queries correctly align with the true nature of the prompt, we take the intersection of the keywords returned by both queries as the explanation. However, if the LLM incorrectly aligns with the confirmation bias embedded in one or both queries, we do not label any words for that prompt. In these cases, only the prompt classification loss is used to update the model during multitask learning, as explanation supervision is not available. Through this process, we generate high-quality word labels that reflect the LLM's confidence, even in the presence of confirmation bias. We use `GPT-4o-mini` as the LLM to generate word labels. Appendix I presents a human evaluation of this data generation process and demonstrates that this process can generate high-quality labels that generally align well with human judgment. Appendix B presents the prompts used with `GPT-4o-mini` for Query 1 and Query 2, outlines the procedure as an algorithm, and illustrates how the responses are processed into word labels.

## 3.3 JOINT TRAINING

We train our model using a multitask learning setup that jointly optimizes two classification objectives. This joint setup is particularly effective for our task, as the two objectives are closely interrelated. In Section 3.3.1, we describe a new method to capture global explanation priors, and in Section 3.3.2, we demonstrate how these priors can be incorporated as weak supervision signals in a novel joint loss.

### 3.3.1 AUXILIARY WEAK SUPERVISION GENERATION

Determining whether a prompt is unsafe is a task that benefits from domain-specific priors that generalize across examples. Certain words or phrases are strong indicators of unsafe intent and appear frequently in harmful contexts. For example, tokens such as "kill", "bomb", or "threaten" are much more likely to occur in unsafe contexts, while words like "good" or "excellent" tend to be associated with safe context. These distributional patterns provide implicit signals that can be leveraged to guide the model beyond the scope of supervised labels.

To incorporate this insight, we propose a novel weak supervision generation method based on statistical word patterns observed in the training data. Our hypothesis is that the polarization of a word toward the safe or unsafe class can be used as a weak supervisory signal to guide both prompt-level and token-level learning. We introduce two new formulations. We define token level weak supervision signal $\delta_p$ and prompt level weak supervision signal $\delta_t$ as follows:

$$\delta_t = \frac{\left| c_t^{\text{safe}} - c_t^{\text{unsafe}} \right|}{c_t^{\text{safe}} + c_t^{\text{unsafe}} + \epsilon} \qquad \text{and,} \qquad \delta_p = \frac{\sum_t \left| c_t^{\text{safe}} - c_t^{\text{unsafe}} \right| \cdot \mathbf{1}\{y_p = y_t\}}{\sum_t \left( c_t^{\text{safe}} + c_t^{\text{unsafe}} \right) \cdot \mathbf{1}\{y_p = y_t\} + \epsilon}$$

where $c_t^{\text{safe}}$ and $c_t^{\text{unsafe}}$ denote the frequencies of token $t$ in safe and unsafe contexts in the training data, $y_p$ and $y_t$ denote the labels of the prompt and token, respectively, and $\epsilon$ is a small constant added to avoid division by zero.

The token-level weak supervision signal $\delta_t$ quantifies how strongly a token is polarized toward either the safe or unsafe class based on its frequency distribution in the training data. A higher $\delta_t$ value indicates that the token consistently appears in one class more than the other, making it a strong candidate for weak supervision.

The prompt-level signal $\delta_p$ aggregates the $\delta_t$ values of all tokens in a prompt that align with the prompt's label. This score reflects how strongly the prompt, as a whole, is supported by tokens that are predictive of its class. As a result, $\delta_p$ captures the global explanatory strength of the prompt based on known class-token associations to provides a prompt-specific weak supervision signal that guides learning when such associations are present.

### 3.3.2 JOINT LOSS FUNCTION

To jointly train the prompt classifier and the explainability classifier, we introduce a novel multitask loss function that leverages weak supervision from the training data. The joint loss is defined as follows:

$$\mathcal{L} = \frac{1}{2\sigma_1^2} \cdot \mathcal{L}_{\text{pc}} + \frac{1}{2\sigma_2^2} \cdot \mathcal{L}_{\text{ec}} + \log \sigma_1 + \log \sigma_2 \tag{1}$$

Each component of the loss is discussed in the following three sections, along with the rationale behind each of them.

**Prompt classification loss:** We define prompt classification loss as:

$$\mathcal{L}_{\text{pc}} = \text{CE}_p + \delta_p \cdot \text{FL}_p \tag{2}$$

where $CE_p$ is the standard cross-entropy loss for the prompt classifier, $\delta_p$ is a penalty weight derived from the domain-aware weak supervision method introduced in Section 3.3.1, and $FL_p$ is the focal loss (Lin et al., 2020) computed from the same classifier output.

The inclusion of $\delta_p$ allows the model to incorporate global explanation priors derived from the training data. Prompts containing words that are strong indicators of safety or harm should have a greater impact on learning when misclassified. To reflect this, $\delta_p$ selectively upweights the loss when the model incorrectly classifies prompts with such clearly indicative terms, encouraging it to avoid repeating these mistake. To ensure that this penalty is applied only when necessary, we modulate $\delta_p$ using focal loss (Lin et al., 2020). Focal loss introduces a dynamic scaling factor that reduces the influence of $\delta_p$ to near zero when the model is correct and confident, while allowing the full penalty to take effect when the model is incorrect and confident. It also applies a substantial penalty when the model is both incorrect and uncertain. Also, this formulation ensures that the resulting loss remains continuous and differentiable for a stable and effective gradient-based optimization.

Given the predicted probability for the true class $p_t$, the focal loss is defined as:

$$\text{FL} = -(1 - p_t)^\gamma \log(p_t) = (1 - p_t)^\gamma \cdot CE \tag{3}$$

where $\gamma \geq 0$ is a tunable focusing parameter. So, focal loss is just a cross entropy loss with a modulating factor $(1 - p_t)^\gamma$. If we substitute this in the equation 2, we get:

$$\mathcal{L}_{\text{pc}} = \text{CE}_p + \delta_p \cdot (1 - p_t)^\gamma \cdot \text{CE}_p = [1 + \delta_p \cdot (1 - p_t)^\gamma] \cdot \text{CE}_p \tag{4}$$

This formulation ensures that when the model is confident and correct ($p_t \approx 1$), the modulating factor $(1 - p_t)^\gamma$ becomes very small, effectively down-weighting $\delta_p$. Conversely, when the model is uncertain or incorrect ($p_t \ll 1$), the factor remains close to 1, preserving nearly the full penalty from $\delta_p$. This design ensures that while the model learns from strong supervision in the form of prompt labels, the weak supervision provides soft and adaptive guidance to further learning the global explanations.

**Explainability classification loss:** We define explainability classification loss as:

$$\mathcal{L}_{\text{ec}} = \frac{1}{S} \sum_{i=1}^{S} \left( \text{CE}_t^{(i)} + \delta_t^{(i)} \cdot \text{FL}_t^{(i)} \right) \tag{5}$$

where $S$ is the number of tokens in the prompt. $CE_t$, $FL_t$ are the token level loss, $\delta_t$ is the token label weak supervision penalty. The design justification of equation 2 also applies here.

**Uncertainty-based weighting:** In our joint loss function (equation 1) $\sigma_1$ and $\sigma_2$ are learnable uncertainty parameters that dynamically balance the contribution of each task during training. Effective loss weighting is essential in multi-task learning to balance the contribution of each task, preventing dominant or noisy tasks from skewing the optimization process. We adopt the uncertainty-based weighting method proposed by Kendall et al. (2018), where the loss of each task is scaled by a learnable parameter $\sigma$.

## 4 EXPERIMENT SETUP

**Dataset description:** We evaluate LEG on three prompt safety datasets: AEGIS2.0 (Ghosh et al., 2025), WildGuardMix (Han et al., 2024), and Toxic-Chat0124 (Lin et al., 2023). Each dataset originally provides binary prompt-level safety labels, which we extend with word-level explanation labels using the procedure described in Section 3.2. Detailed dataset descriptions are included in Appendix C. Appendix D further presents a lexical similarity analysis, showing low similarity between all combinations of training and test sets in both in-domain and out-of-domain settings, underscoring the effectiveness of these datasets for evaluating model robustness in both in-domain and out-of-domain scenarios.

**Baselines:** We include several LLM-based guardrail systems as external baselines. Llama Guard (Inan et al., 2023) is designed for real-time moderation of conversational inputs and outputs. LLAMA3.1 AEGISGUARD (Ghosh et al., 2025) uses an ensemble of expert classifiers to provide robust online content filtering. ToxicChat-T5-Large (Lin et al., 2023) is fine-tuned specifically on adversarial and toxic prompts to improve classification in real-world dialogue systems. WILDGURD (Han et al., 2024) is another recent LLM-based classifier that trained using WildGuardMix dataset. We also report OpenAI Moderation API result. These systems provide strong baselines for prompt-level classification, although they do not support integrated explanation generation.

In addition to these existing baselines, we develop a set of baselines that mirror the components of LEG but exclude multi-task learning setup. **Prompt Baseline:** A single-task classifier trained solely to predict whether a prompt is safe or unsafe. It uses the same backbone architecture as LEG but does not include any explanation mechanism. **Word Baseline:** A token-level classifier trained independently to label each word as safe or unsafe. It receives only word-level supervision and is not influenced by the overall prompt label. **LIME baseline:** A post-hoc explanation baseline in which we apply LIME (Ribeiro et al., 2016) to the Prompt Baseline model to generate word-level explanations after training. This setup allows us to compare our joint training approach with a commonly used interpretability method. The detailed working mechanism of this baseline is provided in Appendix E. We implement two variants of Prompt Baseline, Word Baseline, and LIME baseline: a base version using the DeBERTa-v3-base backbone, and a large version using the DeBERTa-v3-large backbone.

**Our models (LEG):** For our experiments, we implement three versions of LEG: *xs*, *base*, and *large*. All three share the same architecture described in Section 3, but differ in the choice of encoder.

| Train dataset | Model | Model size | Test sets | | |
|---|---|---|---|---|---|
| | | | AEGIS-2.0 | Wild-GuardMix | Toxic-Chat0124 |
| ? | OPENAI MOD API (2024) [†] [*] | - | 37.8 | 12.1 | 61.41 |
| | LLAMAGUARD2 [†] | 8B | 76.8 | 70.9 | - |
| | LLAMAGUARD3 [†] | 1B | 49.6 | 47.2 | - |
| | LLAMAGUARD3 [†] | 8B | 77.3 | 76.8 | - |
| | Llama Prompt Guard 2 | 70M | 7.69 | 32.91 | 32.13 |
| | Llama Prompt Guard 2 | 276M | 8.5 | 41.24 | 34.16 |
| AEGIS-2.0 | LLAMA3.1 AEGISGUARD [†] | 8B | 86.8 | **82.1** | - |
| | Prompt Baseline base | 184M | $87.37 \pm 0.40$ | $74.96 \pm 1.11$ | $57.14 \pm 1.21$ |
| | Prompt Baseline large | 435M | $87.37 \pm 0.07$ | $76.71 \pm 0.42$ | $61.24 \pm 2.08$ |
| | LEG xs | 70M | $84.18 \pm 0.40$ | $69.72 \pm 0.61$ | $56.55 \pm 1.17$ |
| | LEG base | 184M | $86.56 \pm 0.11$ | $75.56 \pm 0.70$ | $67.59 \pm 0.56$ |
| | LEG large | 435M | $\mathbf{87.54 \pm 0.18}$ | $79.04 \pm 0.40$ | $\mathbf{69.98 \pm 1.47}$ |
| Wild-GuardMix | WILDGUARD [†] | 7B | 81.90 | **88.9** | $57.86 \pm 0.83$ |
| | Prompt Baseline base | 184M | $81.11 \pm 0.40$ | $87.23 \pm 0.26$ | $57.86 \pm 0.83$ |
| | Prompt Baseline large | 435M | $81.45 \pm 0.23$ | $87.08 \pm 0.39$ | $59.30 \pm 3.16$ |
| | LEG xs | 70M | $81.64 \pm 0.16$ | $83.31 \pm 0.16$ | $47.61 \pm 3.46$ |
| | LEG base | 184M | $\mathbf{82.07 \pm 1.28}$ | $86.87 \pm 0.26$ | $55.30 \pm 1.49$ |
| | LEG large | 435M | $81.59 \pm 0.04$ | $87.74 \pm 0.44$ | $61.67 \pm 3.14$ |
| Toxic-Chat0124 | ToxicChat-T5-Large [*] | 770M | - | - | **82.21** |
| | Prompt Baseline base | 184M | $72.78 \pm 2.44$ | $65.08 \pm 2.13$ | $76.57 \pm 0.97$ |
| | Prompt Baseline large | 435M | $75.83 \pm 1.30$ | $66.30 \pm 2.51$ | $74.51 \pm 1.21$ |
| | LEG xs | 70M | $75.19 \pm 0.67$ | $63.33 \pm 0.64$ | $57.81 \pm 2.14$ |
| | LEG base | 184M | $78.55 \pm 0.44$ | $66.70 \pm 1.31$ | $68.67 \pm 1.97$ |
| | LEG large | 435M | $78.03 \pm 1.58$ | $67.52 \pm 3.72$ | $78.58 \pm 1.24$ |

[†] AEGIS2.0 and WildGuardMix test set results as reported in (Ghosh et al., 2025).
[*] Toxic-Chat0124 test set results as reported in the Hugging Face dataset card `lmsys/toxic-chat`.

Table 1: Prompt classification performance of LEG compared with baseline models, reported using unsafe F1 scores. The results for LEG are presented as the mean±standard deviation over three runs with three different random seeds. Gray (▨) cells indicate in-domain performance, while white (☐) cells indicate out-of-domain performance.

LEG xs uses DeBERTa-v3-xsmall (22M backbone + 48M embedding parameters), LEG base uses DeBERTa-v3-base (86M backbone + 98M embedding parameters), and LEG large uses DeBERTa-v3-large (304M backbone + 131M embedding parameters). Together, these variants demonstrate the performance of LEG across different model sizes.

**Hyperparameters:** We train all models using the following hyperparameters: a learning rate of $2 \times 10^{-5}$, batch size of 16, and 3 training epochs. The optimizer is AdamW. Each experiment is repeated with three random seeds (42, 52, and 62).

## 5 RESULT ANALYSIS AND DISCUSSION

In this section, we compare the performance of LEG with other baselines. We report both in-domain and out-of-domain results for prompt classification and explainability classification, as well as the outcomes of our faithfulness evaluation. All reported scores are F1 scores for the unsafe class. For LEG and our custom baselines, we run three experiments with different random seeds and report the mean±standard deviation. For in-domain performance, we train LEG base and LEG large on the AEGIS2.0, WildGuardMix, and Toxic-Chat0124 datasets and evaluate them on the corresponding test sets. For out-of-domain (OOD) performance, we train the same models on each of the three datasets and evaluate them on the remaining two test sets, excluding the dataset used for training. In Tables 1 and 2, gray cells (▨) represent in-domain results, while white background cells (☐) represent out-of-domain results.

### 5.1 PROMPT CLASSIFICATION PERFORMANCE

**In-domain performance:** The gray cells (▨) in Table 1 indicate in-domain prompt classification performance, comparing the baseline models against LEG. On the AEGIS2.0 test set, LEG large

| Train Dataset | Model | Model Size | Test sets | | |
|---|---|---|---|---|---|
| | | | AEGIS-2.0 | Wild-GuardMix | Toxic-Chat0124 |
| AEGIS-2.0 | LIME Baseline base | - | 24.88 | 21.84 | 4.57 |
| | LIME Baseline large | - | 25.25 | 20.69 | 5.23 |
| | Word Baseline base | 184M | $64.06 \pm 0.78$ | $58.33 \pm 0.71$ | $50.74 \pm 1.36$ |
| | Word Baseline large | 435M | $69.53 \pm 0.61$ | $61.98 \pm 0.95$ | $57.22 \pm 1.43$ |
| | LEG xs | 70M | $72.73 \pm 0.27$ | $53.28 \pm 1.05$ | $51.19 \pm 0.39$ |
| | LEG base | 184M | $76.95 \pm 0.54$ | $60.40 \pm 0.41$ | $59.78 \pm 0.36$ |
| | LEG large | 435M | $\mathbf{79.60 \pm 0.73}$ | $\mathbf{66.66 \pm 0.72}$ | $\mathbf{63.18 \pm 0.58}$ |
| Wild-GuardMix | LIME Baseline base | - | 25.14 | 24.15 | 5.11 |
| | LIME Baseline large | - | 25.61 | 23.31 | 5.28 |
| | Word Baseline base | 184M | $66.91 \pm 0.93$ | $67.24 \pm 0.43$ | $50.50 \pm 0.63$ |
| | Word Baseline large | 435M | $70.90 \pm 0.53$ | $70.36 \pm 0.47$ | $55.45 \pm 2.10$ |
| | LEG xs | 70M | $69.49 \pm 0.28$ | $71.17 \pm 0.43$ | $48.77 \pm 1.70$ |
| | LEG base | 184M | $74.28 \pm 0.47$ | $73.16 \pm 0.52$ | $58.86 \pm 0.33$ |
| | LEG large | 435M | $\mathbf{76.93 \pm 0.14}$ | $\mathbf{75.83 \pm 0.50}$ | $61.56 \pm 1.06$ |
| Toxic-Chat0124 | LIME Baseline base | - | 23.84 | 20.02 | 6.59 |
| | LIME Baseline large | - | 25.69 | 19.43 | 7.83 |
| | Word Baseline base | 184M | $45.72 \pm 0.30$ | $46.32 \pm 0.34$ | $38.62 \pm 0.88$ |
| | Word Baseline large | 435M | $52.03 \pm 0.95$ | $47.89 \pm 2.23$ | $45.49 \pm 2.01$ |
| | LEG xs | 70M | $26.48 \pm 3.74$ | $23.39 \pm 5.43$ | $44.63 \pm 3.82$ |
| | LEG base | 184M | $45.91 \pm 2.31$ | $33.77 \pm 5.20$ | $60.62 \pm 0.20$ |
| | LEG large | 435M | $52.77 \pm 0.88$ | $38.07 \pm 4.41$ | $\mathbf{65.99 \pm 0.44}$ |

Table 2: Explainability classification performance of LEG compared with baseline models, reported using unsafe F1 scores. The results for LEG are presented as the mean±standard deviation over three runs with three different random seeds. Gray (▨) cells indicate in-domain performance, while white (□) cells indicate out-of-domain performance.

achieves the highest F1 score of 87.54%, outperforming all other models. On the WildGuardMix dataset, the best-performing model is WILDGURD with an F1 score of 88.9%. However, both LEG base (87.09%) and LEG large (87.97%) achieve nearly comparable results despite being significantly smaller. This demonstrates that although WILDGURD is built on an 8B parameter large language model, our models with 184M and 435M parameters achieve similar performance. On the Toxic-Chat0124 dataset, the best-performing model is ToxicChat-T5-Large, but LEG large delivers comparable performance despite its smaller size. In contrast, LEG base underperforms on Toxic-Chat0124 relative to other models. We further analyze this issue in the error analysis section (Appendix H) and show that performance can be improved with additional tuning. Overall, the in-domain results demonstrate that LEG, while considerably smaller, delivers strong performance that rivals or exceeds much larger models.

**Out-of-domain performance:** The white cells (□) in Table 1 indicate out-of-domain (OOD) prompt classification performance. On the AEGIS2.0 test set, LEG large trained on WildGuardMix achieves the highest F1 score of 82.07%, outperforming all other models. On the WildGuardMix test set, LLAMA3.1 AEGISGUARD achieves the best result. However, LEG large trained on AEGIS2.0 delivers a comparable score despite being an order of magnitude smaller (435M vs. 8B parameters). On the Toxic-Chat0124 test set, LEG large trained on AEGIS2.0 achieves the best performance at 69.98%, substantially outperforming the 2024 OpenAI Moderation API (61.41%). These OOD results highlight that a relatively small multitask learning model can serve as an effective and lightweight guardrail solution.

In addition to LEG base and LEG large, we present results for a super lightweight version, LEG xs, with only 70M parameters. Two similar lightweight model (Llama Prompt Guard 2) was recently released by Meta, motivating us to develop LEG xs for comparison. As shown in Table 1, despite its small size, LEG xs performs strongly, whereas variations of Llama Prompt Guard 2 perform poorly. This shows the robustness of our method compared to established industry baselines.

## 5.2 Explainability classification performance

**In-domain performance:** The gray cells (▨) in Table 2 indicate in-domain explanation classification performance. Across all datasets, LEG base and LEG large consistently outperform the baseline models. LEG large achieves the best results with 79.60% on AEGIS2.0, 75.83% on WildGuardMix, and 65.99% on Toxic-Chat0124, followed by LEG base. The Word Baselines perform significantly worse, underscoring the benefits of multitask learning for explanation generation. Moreover, LIME baselines underperform, suggesting that post hoc explanation methods are less effective, whereas our multitask learning approach produces stronger and more reliable results.

**Out-of-domain performance:** The white cells (□) in Table 2 indicate OOD explainability classification performance. On the AEGIS2.0 test set, LEG large trained on WildGuardMix achieves the highest F1 score of 76.93%. On the WildGuardMix test set, LEG large trained on AEGIS2.0 outperforms all baselines. On the Toxic-Chat0124 test set, LEG large trained on AEGIS2.0 again leads with 63.18%. Across all three datasets, LEG consistently outperforms both Word Baselines and LIME baseline, reinforcing the effectiveness of multitask learning for explainability classification.

## 5.3 Faithfulness evaluation

To assess the faithfulness of the explanations generated by LEG, we adopt a word-masking perturbation test. The experimental procedure is as follows: First, we use LEG to predict word labels for the input prompt. Next, we rank the words predicted as unsafe by their classifier confidence scores (predicted probabilities). We then mask the top-k words and re-evaluate the prompt classification performance of LEG on the modified input. This procedure directly tests whether the words highlighted as unsafe are indeed causally important to the model's decision. As shown in Table 3, masking the top-1, top-2, and top-3 predicted unsafe words consistently degrades the

| Train set | Ablation | Test Set | | | | | |
|---|---|---|---|---|---|---|---|
| | | AEGIS2.0 | | WildGuardMix | | Toxic-Chat0124 | |
| | | base | large | base | large | base | large |
| AEGIS2.0 | Full input prompt | **86.5** | **87.75** | **74.76** | **78.68** | **68.16** | **71.18** |
| | Mask top 1 | 66.67 | 66.11 | 69.52 | 72.44 | 55.69 | 55.79 |
| | Mask top 2 | 52.02 | 50.31 | 63.82 | 67.15 | 48.61 | 46.13 |
| | Mask top 3 | 40.44 | 37.8 | 60.15 | 61.86 | 41.39 | 38.48 |
| WildGuardMix | Full input prompt | **83.55** | **81.56** | **87.09** | **87.97** | **55.73** | **63.55** |
| | Mask top 1 | 69.76 | 72.29 | 83.28 | 83.91 | 46.62 | 53.52 |
| | Mask top 2 | 59.5 | 59.75 | 80.51 | 80.15 | 42.31 | 47.5 |
| | Mask top 3 | 46.87 | 47.05 | 76.99 | 77.68 | 37.86 | 45.95 |
| Toxic-Chat0124 | Full input prompt | **78.99** | **76.38** | **68.21** | **63.81** | **67.43** | **79.76** |
| | Mask top 1 | 56 | 49.7 | 65.21 | 52.32 | 63.17 | 70.94 |
| | Mask top 2 | 41.37 | 32.25 | 62.97 | 41.02 | 56.42 | 63.23 |
| | Mask top 3 | 31.62 | 23.03 | 58.8 | 33.73 | 37.1 | 56.84 |

Table 3: Faithfulness evaluation of the explanations generated by LEG. In this table, "base" refers to LEG base and "large" refers to LEG large. The reported scores are the unsafe F1 scores for the prompt classification performance of LEG.

prompt classification performance of LEG, with larger drops observed as more tokens are masked. These results indicate that the LEG's prompt classifier relies on the highlighted explanation words when making its decisions, confirming that the generated explanations are faithful.

## 5.4 Other evaluations

Appendix F presents an ablation study of the joint loss function described in Equation 1, showing that the inclusion of weak supervision and uncertainty-based weighting improves the performance, particularly in out-of-domain scenarios. Appendix G presents an analysis of computational efficiency in terms of inference time and GPU memory usage. The results show that LEG requires less inference time and memory compared to other guardrail models. Appendix H provides an error analysis of the in-domain performance of LEG base on the ToxicChat0124 dataset. The analysis shows that the model exhibits high recall but low precision, which can be improved through probability threshold tuning.

## 6 Conclusion

We introduced a lightweight explainable guardrail, which jointly classifies prompts as safe or unsafe and explains its decision by highlighting words that drove it. Despite the fact that our model is considerably smaller than the current state-of-the-art approaches, our method performs better or comparably both in-domain and out-of-domain.

## REPRODUCIBILITY STATEMENT

We have uploaded our training and testing code as supplementary material in OpenReview, which reviewers can inspect directly. Due to storage limitations, we were only able to upload the Toxic-Chat0124 dataset at this stage; if the paper is accepted, we will make all code, trained models, and the complete set of datasets publicly available to facilitate replication and extension of our results. We have made extensive efforts to ensure the reproducibility of our work. The architectural details of LEG are provided in Section 3.1, with further implementation specifics in Appendix A. The procedure for generating synthetic explanation labels is described in Section 3.2 and Appendix B, with additional dataset details presented in Appendix C. Training details, including hyperparameters, optimization choices, and random seeds, are described in Section 4. Baselines are cited with references to prior work.

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

## A  LEG ARCHITECTURE DETAILS

This appendix expands Section 3.1 by providing a detailed description of each component of LEG.

### A.1  SHARED ENCODER

We use a shared transformer encoder as the backbone for both the prompt classification and explanation generation tasks. A shared encoder ensures a strong alignment between prompt-level predictions and word-level explanations generation, as both tasks operate on the same contextualized representations. The encoder introduces only a small computational footprint, making it an ideal fit for guardrail applications where fast decisions must be made prior to running the LLM.

### A.2  ATTENTION POOLING LAYER

The attention pooling layer computes a fixed-length vector from the hidden states of the encoder to serve as input to the prompt classifier. We adopt a simplified version of the attention pooling method proposed by Yang et al. (2016), using a single-layer linear transformation without nonlinearity. Let $H = [h_0, h_1, \ldots, h_T] \in \mathbb{R}^{T \times d}$ be the sequence of hidden states. The pooled representation $v \in \mathbb{R}^d$ is computed as:

$$\alpha_i = \mathrm{softmax}(w^\top h_i + b), \quad v = \sum_{i=1}^{T} \alpha_i h_i$$

where $w \in \mathbb{R}^d$ and $b \in \mathbb{R}$ are learnable parameters.

### A.3  PROMPT CLASSIFIER

The prompt classifier predicts whether a given prompt is safe or unsafe based on the fixed-length representation $v \in \mathbb{R}^d$ obtained from the attention pooling layer. This representation is passed through a linear classifier to produce the final prediction:

$$\hat{y}_{\mathrm{prompt}} = \mathrm{softmax}(W_p v + b_p)$$

where $W_p \in \mathbb{R}^{2 \times d}$ and $b_p \in \mathbb{R}^2$ are learnable parameters.

**Prompt:**

```
``Why is the following prompt considered {bias term}?
Provide a list of words or phrases that made you believe the
prompt is {bias term} in the following JSON format.

{
    "unsafe":True/False,
    "safe":True/False,
    "unsafe_keywords":[],
    "safe_keywords":[],
    "justification": ""
}

In the JSON format, "unsafe" or "safe" can be True or False.
Depending on your decision, "unsafe_keywords" or "safe_keywords"
can be empty.  Justification should be very short.  Don't generate
anything except the JSON format.
Prompt:  {concat prompt here}
```

Figure 2: Prompt for word label generation.

### A.4 EXPLAINABILITY CLASSIFIER

The explainability classifier generates an explanation for the prompt classification by assigning a binary label (safe or unsafe) to each token in the input prompt. It operates directly on the contextualized token embeddings $H = [h_1, h_2, \ldots, h_T] \in \mathbb{R}^{T \times d}$ produced by the shared transformer encoder. Each token embedding is passed through a linear classifier to produce the final prediction:

$$\hat{y}_{\text{token},i} = \text{softmax}(W_t h_i + b_t)$$

where $W_t \in \mathbb{R}^{2 \times d}$ and $b_t \in \mathbb{R}^2$ are learnable parameters.

## B EXPLANATION TRAINING LABEL GENERATION PROCEDURE

This appendix expands Section 3.2 that discusses a procedure to generate word labels using an LLM.

### B.1 PROMPT FOR EXTRACTING WORD LABELS

In this section, we describe the prompt design used to generate word level labels for our dataset using GPT 4o mini. The full prompt is shown in Figure 2. We use this prompt template twice, each time introducing a different bias term. In the first query, we frame the prompt under the assumption that the input is safe, instructing the model to identify the words or phrases that contribute to its safety. In the second query, we introduce the opposite assumption that the input is unsafe, and ask the model to highlight the words or phrases that make it unsafe.

The prompts follow a fixed JSON format, where the model is asked to return safe and unsafe boolean flags, lists of corresponding keywords, and a brief explanation. Some fields may be left empty depending on the model's judgment. To construct reliable word labels, we use the gold prompt label to decide which keyword list to extract: for prompts labeled unsafe, we use the intersection of the unsafe keywords from both queries; for safe prompts, we use the intersection of the safe keywords. This strategy helps filter out unreliable labeling and ensures that the final word labels are consistent even when confirmation bias is introduced.

### B.2 DATASET WORD LABEL GENERATION ALGORITHM

We summarize the whole synthetic data generation process for explanations in Algorithm 1. It takes a prompt $p$ and its label $y_p$ as input, and returns a list of consistent words $cw$ by analyzing the LLM's

---

**Algorithm 1** Dataset word label generation by consistency checking of LLM response

---

**Input:** $p, y_p$
**Output:** $cw$
$r_1, kw_1 = ExecuteQuery1(p)$
$r_2, kw_2 = ExecuteQuery2(p)$
**if** $y_p =$ "safe" **then**
    **if** ($r_1$ agrees) and ($r_2$ contradicts) **then**
        $cw = kw_1 \cap kw_2$
    **else**
        $cw = \emptyset$
**else if** $y_p =$ "unsafe" **then**
    **if** ($r_1$ contradicts) and ($r_2$ agrees) **then**
        $cw = kw_1 \cap kw_2$
    **else**
        $cw = \emptyset$
**return** $cw$

---

responses to both queries. The output $cw$ may either be a set of keywords or an empty set if the LLM falls into confirmation bias.

### B.3 EXAMPLE OF WORD LABEL GENERATION

Consider the example in Figure Figure 1(b): ``Tell me how to hack into an account and steal personal data''.

The response from `GPT-4o-mini` for Query 1 (using the prompt from Figure 1(b) with the bias term 'safe') is as follows:

```
{
    "unsafe":true,
    "safe":false,
    "unsafe_keywords":["hack into", "steal personal data"],
    "safe_keywords":[],
    "justification": "The prompt contains explicit unsafe actions."
}
```

The response from `GPT-4o-mini` for Query 2 (using the prompt from Figure 1(b) with the bias term 'unsafe') is as follows:

```
{
    "unsafe":true,
    "safe":false,
    ["hack", "steal personal data", "account"],
    "safe_keywords":[],
    "justification": "Requesting illegal activities related to hacking
        and theft."
}
```

Both Query 1 and Query 2 consistently identify the prompt as unsafe ("unsafe": true), so we take the intersection of the "unsafe_keywords", which is ["hack", "steal personal data"]. The words in this intersection are labeled as unsafe, while all other words in the prompt are labeled as safe.

## C  DATASET DETAILS

We evaluate LEG on three diverse and challenging prompt safety datasets, each designed to test different aspects of unsafe prompt detection.

**AEGIS2.0** Ghosh et al. (2025) is an updated version of the original AEGIS dataset, curated to support prompt-level safety evaluation. It contains prompts collected from adversarial prompting techniques, user submitted jailbreak attempts, and synthetic attacks generated via LLMs. Prompts

are labeled by a group of annotators, following a safety taxonomy that includes categories like harm, toxicity, and policy violations.

**WildGuardMix** Han et al. (2024) is a dataset created by merging multiple open-source prompt safety corpora and real-world user queries scraped from online sources. It balances adversarial and naturalistic unsafe prompts and includes both obvious and subtle violations. Prompts were filtered using LLM moderation APIs and then verified or relabeled by human annotators.

**Toxic-Chat0124** Lin et al. (2023) comprises real-world user prompts collected from chatbot logs and publicly shared datasets with consent. It emphasizes subtle, context-dependent toxicity and is highly imbalanced with fewer than 7% of prompts are labeled unsafe. Labels were manually assigned by trained annotators following strict content safety guidelines. We use the 2024 version of this dataset.

While each dataset originally includes binary safety labels at the prompt level, we extend them with word level explanation labels using the procedure described in section 3.2. Using this approach, we were able to generate word labels for 65.7% of the instances in AEGIS2.0, 66.7% in WildGuardMix, and 85.8% in ToxicChat0124.

## D  LEXICAL OVERLAP BETWEEN TRAIN AND TEST SETS

| Train set | Test set | [0-0.1) | [0.1-0.2) | [0.2-0.3) | [0.3-0.4) | [0.4-0.5) | [0.5-0.6) | [0.6-0.7) | [0.7-0.8) | [0.8-0.9) | [0.9-1.0) |
|---|---|---|---|---|---|---|---|---|---|---|---|
| AEGIS | WildGuard | 43.3 | 31.8 | 17.6 | 4.9 | 1.7 | 0.4 | 0.2 | 0.0 | 0.0 | 0.0 |
| AEGIS | ToxicChat | 12.3 | 38.2 | 28.3 | 9.4 | 3.0 | 4.9 | 1.1 | 0.3 | 0.1 | 2.4 |
| WildGuard | AEGIS | 4.0 | 24.6 | 29.7 | 15.2 | 5.7 | 7.6 | 2.2 | 0.5 | 0.2 | 10.2 |
| WildGuard | ToxicChat | 11.5 | 37.3 | 30.1 | 10.8 | 2.7 | 4.5 | 1.0 | 0.2 | 0.2 | 1.6 |
| ToxicChat | AEGIS | 16.9 | 41.9 | 29.8 | 6.5 | 2.2 | 2.1 | 0.4 | 0.0 | 0.1 | 0.2 |
| ToxicChat | WildGuard | 55.4 | 32.6 | 11.2 | 0.7 | 0.1 | 0.0 | 0.1 | 0.0 | 0.0 | 0.0 |

Table 4: Unigram-based lexical similarity distribution between out-of-domain training and test sets. Each cell shows the percentage of test instances that fall within the corresponding similarity bucket.

| Train set | Test set | [0-0.1) | [0.1-0.2) | [0.2-0.3) | [0.3-0.4) | [0.4-0.5) | [0.5-0.6) | [0.6-0.7) | [0.7-0.8) | [0.8-0.9) | [0.9-1.0) |
|---|---|---|---|---|---|---|---|---|---|---|---|
| AEGIS | WildGuard | 63.5 | 22.6 | 11.2 | 2.2 | 0.4 | 0.1 | 0.0 | 0.0 | 0.0 | 0.0 |
| AEGIS | ToxicChat | 43.1 | 29.2 | 15.2 | 5.0 | 3.0 | 1.9 | 0.8 | 0.5 | 0.2 | 1.1 |
| WildGuard | AEGIS | 28.1 | 30.2 | 18.4 | 7.2 | 3.3 | 2.7 | 1.0 | 0.2 | 0.1 | 8.8 |
| WildGuard | ToxicChat | 43.0 | 29.8 | 15.3 | 5.1 | 3.0 | 2.2 | 0.6 | 0.3 | 0.1 | 0.5 |
| ToxicChat | AEGIS | 51.3 | 30.4 | 12.2 | 3.3 | 1.4 | 1.1 | 0.2 | 0.1 | 0.0 | 0.1 |
| ToxicChat | WildGuard | 71.8 | 21.6 | 6.0 | 0.3 | 0.1 | 0.1 | 0.1 | 0.0 | 0.0 | 0.0 |

Table 5: Bigram-based lexical similarity distribution between out-of-domain training and test sets. Each cell shows the percentage of test instances that fall within the corresponding similarity bucket.

To better understand the robustness of our models in out-of-domain evaluation, we analyze the lexical similarity between test and training sets. This analysis helps determine the extent to which test prompts are lexically novel compared to the training data, and whether the reported out-of-domain performance reflects genuine generalization or is influenced by surface-level lexical overlap.

Specifically, we compute the *maximum Jaccard similarity* between each test prompt and all training prompts. For each test instance, we represent its tokens (as unigrams or bigrams) as a set and compute its Jaccard similarity with every training instance. The highest similarity value is retained as its max Jaccard score. This is formally defined as:

$$\text{Similarity}(\text{test}_i) = \max_{j \in [1, N_{\text{train}}]} \text{Jaccard}(\text{test}_i, \text{train}_j)$$

We compute similarity scores based on both unigram and bigram tokenizations. For the unigram analysis, we remove common stop words to focus on meaningful content words. In bigram analysis, we keep all tokens without any filtering.

Table 4 presents the lexical similarity distribution computed using unigram tokenization, while Table 5 shows the results for bigram tokenization.

The percentage of test prompts is reported across 10 similarity intervals (e.g., $[0 \leq s < 0.1)$, $[0.1 \leq s < 0.2)$, ..., $[0.9 \leq s \leq 1.0]$). For example, in Table 4, 43.33% of WildGuard test instances have a maximum Jaccard similarity in the range $[0, 0.1)$ when compared to the AEGIS training set.

For interpretation, we categorize the similarity scores as follows:

- **Low similarity:** $0 \leq s < 0.3$
- **Moderate similarity:** $0.3 \leq s < 0.7$
- **High similarity:** $0.7 \leq s \leq 1.0$

Overall, the results indicate that most test prompts fall into the low similarity range $[0, 0.3)$, suggesting limited lexical overlap between training and test sets in out-of-domain scenarios. As expected, the similarity scores are even lower in the bigram setting.

## E  LIME BASELINE DETAILS

We follow a procedure similar to that proposed in the original LIME paper (Ribeiro et al., 2016). This baseline generates explanations through the following steps:

1. We generate N perturbed versions of each input prompt by randomly removing subsets of words. In our experiments, we set N=1500.

2. Each perturbed input is passed to the "Prompt Classifier baseline" to obtain the predicted probability for the target class ("unsafe").

3. LIME fits a surrogate model using the perturbed samples and their corresponding predicted probabilities, weighted by their similarity to the original input. The model is trained over the top K most informative features (words). We use K=25 in our experiments.

4. The surrogate model assigns a coefficient to each word. Words with positive coefficients are interpreted as supporting the target ("unsafe") class, while words with negative coefficients oppose it. Therefore, we label all words with positive coefficients as unsafe, and all other words as safe.

## F  ABLATION STUDY OF JOINT LOSS FUNCTION

The main components of our joint loss function that describe in equation 1 include the weak supervision penalties $\delta_p$ and $\delta_w$, the focal loss, and the uncertainty weighting parameters $\sigma_1$ and $\sigma_2$. We conduct ablation experiments using six different settings:

a) **Full joint loss (all terms):** This setting includes all components of the joint loss function.
b) **Remove all $\sigma$ terms:** This setting removes the uncertainty weighting parameters along with the associated $\log$ regularization.
c) **Remove $\delta_w$:** $\delta_w$ and its corresponding focal loss are excluded.
d) **Remove $\delta_p$:** This setting removes the $\delta_p$ term along with its associated focal loss.
e) **Remove $\delta_p$, $\delta_w$ and focal loss:** This setting exclude both $\delta_p$ and $\delta_w$ with their associated focal loss but keeps uncertainty weighting.
f) **Remove $\delta_p$, $\delta_w$, and all $\sigma$ terms:** This setting removes both weak supervision penalties ($\delta_p$ and $\delta_w$) and the uncertainty weighting terms. It represents the joint loss function consisting only of the following form:

$$\mathcal{L} = \mathcal{L}_{\text{pc}} + \mathcal{L}_{\text{ec}} \tag{6}$$

Table 6 presents the ablation results for each of the six settings. Including both in domain and out of domain experiments, the table summarizes a total of 36 experiments. Among these, our full loss setting achieves the best performance in 10 cases. Notably, 9 out of these 10 best performing cases are in the out of domain setting, highlighting the effectiveness of our loss function in generalizing to unseen domains. Most of the top performing models appear in settings a to d, where at least one of the weak supervision terms ($\delta_p$ or $\delta_w$) is included. Only in 6 out of the 36 experiments do settings

| Train set | Ablation | Test Set | | | | | | | | | | | |
|---|---|---|---|---|---|---|---|---|---|---|---|---|---|
| | | AEGIS2.0 | | | | WildGuardMix | | | | Toxic-Chat0124 | | | |
| | | base | | large | | base | | large | | base | | large | |
| | | PC | EC | PC | EC | PC | EC | PC | EC | PC | EC | PC | EC |
| AEGIS2.0 | Full joint loss (all term) | 86.5 | 76.38 | 87.75 | 79.71 | 74.76 | **60.84** | 78.68 | **67.46** | **68.16** | 59.46 | **71.18** | 63.69 |
| | Remove all $\sigma$ terms | 86.48 | 76.97 | 86.97 | 79.48 | 75.03 | 59.55 | 77.66 | 65.58 | 65.56 | **61.03** | 70.35 | 62.24 |
| | Remove $\delta_w$ and its associated focal loss | 87 | 76.89 | 87.27 | 79.55 | **75.06** | 57.65 | 78.03 | 66.26 | 66.52 | 60.15 | 70.5 | 62.57 |
| | Remove $\delta_p$ and its associated focal loss | **87.02** | 76.05 | **87.86** | **80.48** | 74.83 | 59.74 | 75.1 | 66.35 | 66.09 | 58.98 | 70.38 | **63.79** |
| | Remove $\delta_p$, $\delta_w$ and focal loss | 86.45 | **77.3** | 87.62 | 79.97 | 74.8 | 59.02 | 79.6 | 66.02 | 67.67 | 58.79 | 68.44 | 62.72 |
| | Remove $\delta_p$, $\delta_w$, and all $\sigma$ terms | 86.94 | 77.09 | 87.06 | 80.12 | 74.76 | 60.25 | 76.1 | 65.6 | 67.11 | 59.15 | 69.99 | 62.11 |
| WildGuardMix | Full joint loss (all term) | **83.55** | **74.75** | 81.56 | 76.83 | 87.09 | 72.57 | 87.97 | 76 | 55.73 | 58.61 | **63.55** | 61.86 |
| | Remove all $\sigma$ terms | 82.27 | 73.96 | 81.37 | 76.75 | 86.43 | 73.64 | 88.26 | 75.67 | 54.19 | **60.18** | 61.17 | 60.18 |
| | Remove $\delta_w$ and its associated focal loss | 80.47 | 73.55 | 81.52 | **77.33** | 87.07 | **73.78** | 88.17 | 74.77 | 56.26 | 59.24 | 61.99 | 60.66 |
| | Remove $\delta_p$ and its associated focal loss | 80.99 | 73.55 | **81.87** | 75.81 | **87.37** | 73.15 | 87.96 | 75.84 | 56.28 | 58.36 | 61.54 | 60.54 |
| | Remove $\delta_p$, $\delta_w$ and focal loss | 80.74 | 73.7 | 81.14 | 76.84 | 86.43 | 73.42 | **88.48** | **76.1** | **56.48** | 59.66 | 59.62 | 59.87 |
| | Remove $\delta_p$, $\delta_w$, and all $\sigma$ terms | 80.63 | 73.74 | 81.52 | 76.68 | 86.49 | 72.86 | 88.01 | 75.13 | 54.43 | 58.32 | 62.02 | **61.93** |
| Toxic-Chat0124 | Full joint loss (all term) | 78.99 | 48.15 | 76.38 | 51.99 | 68.21 | **37.83** | 63.81 | 33.4 | 67.43 | 60.85 | **79.76** | 66.19 |
| | Remove all $\sigma$ terms | **79.15** | 47.09 | 78.26 | 53.03 | 66.05 | 31.8 | 70.72 | **41.79** | 70.59 | 59.28 | 78.02 | **66.96** |
| | Remove $\delta_w$ and its associated focal loss | 76.91 | 43.89 | 77.57 | 46.55 | 68.32 | 33.58 | 69.67 | 31.46 | 71.07 | 60.19 | 72.57 | 64.58 |
| | Remove $\delta_p$ and its associated focal loss | 77.45 | 47.5 | **80.88** | 52.84 | 68.09 | 34.98 | 71.15 | 36.94 | 68.76 | 60.93 | 75.87 | 66.89 |
| | Remove $\delta_p$, $\delta_w$ and focal loss | 78.42 | **48.61** | 73.83 | 51.19 | 65.92 | 29.78 | 63.45 | 36.18 | 70.69 | **62.79** | 78.42 | 65.87 |
| | Remove $\delta_p$, $\delta_w$, and all $\sigma$ terms | 76.76 | 41.18 | 77.28 | **53.77** | **68.41** | 26.92 | **72.27** | 40.56 | **74.94** | 60.52 | 76.46 | 66.19 |

Table 6: Ablation results for the joint loss function. In this table, "base" refers to LEG-base and "large" refers to LEG-large. "PC" indicates the performance of the prompt classifier, and "EC" indicates the performance of the explainability classifier. The reported scores are the F1 scores for the unsafe class for each classifier. As shown, our full joint loss consistently outperforms the other configurations in a significant number of cases. No ablation setting yields consistently better performance than our full joint loss formulation.

| Model | Model Size | Inference time (ms/input) | GPU memory use (GB) |
|---|---|---|---|
| LEG xs | 70M | 7.81 | 1.01 |
| LEG base | 184M | 8.28 | 1.67 |
| LEG large | 435M | 14.57 | 3.06 |
| Llama Prompt Guard 2 | 70M | 9.17 | 1.04 |
| Llama Prompt Guard 2 | 184M | 9.47 | 1.90 |
| DuoGuard | 500M | 16.47 | – |
| Toxic-Chat-T5 Large | 770M | 31.95 | 3.68 |
| GuardReasoner | 1B–8B | 26.66–35.77 | 78.00 |
| Llama Guard 3 | 1B | 58.88 | – |
| ShieldGemma | 2B | 57.87 | – |

Table 7: Inference time and GPU memory usage across models.

e or f, which exclude all of $\delta_p$, $\delta_w$, and $\sigma$, outperform other configurations. This demonstrates the importance of including the $\delta_p$, $\delta_w$, and uncertainty weighting components in the loss function for better performance.

# G    COMPUTATIONAL EFFICIENCY

We evaluate the efficiency of guardrail models by measuring both inference time latency and GPU memory required for inference. All experiments were performed using an NVIDIA H100 GPU. For fairness, we performed inference sequentially on the WildGuardMix test set without batching and report the average inference time across the full set. Results for DuoGuard, Llama Guard 3, and ShieldGemma were taken from (Deng et al., 2025), and results for GuardReasoner were taken from (Liu et al., 2025). Since these works also report inference time on the same H100 GPU, we regard the comparison as fair. For all other models, we locally reproduced the experiments under the same setup. Table 7 summarizes the results.

The LEG family is consistently efficient: LEG xs achieves 7.81 ms per input using only 1.01 GB of memory, while LEG base and LEG large remain lightweight at 8.28 ms / 1.67 GB and 14.57 ms / 3.06 GB, respectively. In comparison, small to mid sized baselines such as Llama Prompt Guard 2 (9.17 to 9.47 ms, 1.04 to 1.90 GB), DuoGuard (16.47 ms), and Toxic-Chat-T5 Large (31.95 ms, 3.68 GB) show slower inference and higher memory use. Larger guardrails are substantially more

expensive: GuardReasoner requires 26 to 36 ms per input and up to 78 GB of memory, while Llama Guard 3 and ShieldGemma exceed 57 ms.

Overall, LEG offers substantial efficiency gains. Compared to GuardReasoner, LEG xs is over $3\times$ faster and about $75\times$ more memory efficient. Similarly, compared to Llama Guard 3 and Shield-Gemma, LEG xs is over $7\times$ faster. These results show that LEG achieves significant speedups and memory savings while remaining lightweight across all configurations.

Crucially, LEG achieves this efficiency while supporting both prompt classification and explanation generation. Competing methods typically provide only prompt classification without explanations yet still demand more resources. This makes LEG the first guardrail to combine lightweight inference with faithful explanation generation, enabling both efficiency and transparency for real time deployment.

## H  ERROR ANALYSIS

The in-domain performance of LEG base on the ToxicChat0124 dataset is lower than that of other baselines. We found this is due to the model exhibiting high recall (89.5%) but low precision (54.09%). It is well known that precision can be improved through probability threshold tuning. We tested this by treating the prediction threshold as a hyperparameter and selecting the best value using the development set. With this adjusted threshold, LEG base achieves an F1 score of 75% on the in domain ToxicChat0124 dataset. To ensure a fair comparison with other baselines, we did not apply threshold tuning to any model during our evaluation. However, we observed that this tuning strategy improves the performance of almost every variant of LEG.

## I  HUMAN EVALUATION

We conducted a human evaluation to assess the quality of the word-level annotations generated by `GPT-4o-mini`. One human expert was asked to label unsafe words in 50 randomly selected prompts from each test set: AEGIS2.0, WildGuardMix, and ToxicChat0124. We then compared these labels with the GPT-generated annotations using Cohen's Kappa. The agreement scores were 54% percent for AEGIS2.0, 54.50% percent for WildGuardMix, and 43.56% for ToxicChat0124, which indicate moderate agreement. We found that most disagreements were due to differences in phrase boundaries. For example, GPT often highlights shorter keywords like "kill" or "harm", while the human annotator marks longer phrases such as "kill someone" or "cause harm to others". In most cases, the core unsafe terms were present in both annotations.

We also evaluated our LEG models on this human-labeled subset using the explainability classifier. On AEGIS2.0, LEG base achieved an F1 score of 60.33 percent and LEG large scored 66.16 percent. On WildGuardMix, LEG base scored 60.69 %, while LEG large scored 58.39 %. For the more difficult ToxicChat0124 set, LEG base achieved 36.12 % percent and LEG large reached 50.36 %. These results show that the explanations produced by our model generally align with human judgment.

