# OpenReview forum: "A Lightweight Explainable Guardrail for Prompt Safety"
_ICLR.cc/2026/Conference — ICLR 2026 Conference Withdrawn Submission_

### Official Review · Reviewer_GwUv · 2025-10-23

**Soundness:** 3
**Presentation:** 3
**Contribution:** 3
**Rating:** 2
**Confidence:** 4

**Summary:**

This paper proposes LEG, a lightweight, explainable guardrail for unsafe-prompt detection. Unlike retraining via RLHF or DPO, LEG is modular, LLM-agnostic, and low-latency. It uses multi-task learning with a shared encoder: one head classifies prompts as safe or unsafe, another highlights words that justify the decision. Because token-level labels are scarce, the authors synthesize explanations with a strategy designed to counter confirmation bias. Training employs a joint loss encoding global explanation priors, combining cross-entropy and focal losses with uncertainty weighting. Experiments on three datasets report SOTA or near-SOTA classification and faithful explanations, plus efficiency gains over heavier guardrails in practice.

**Strengths:**

The paper emphasizes the necessity of faithfulness for unsafe-prompt detectors and explicitly evaluates whether token-level rationales truly support the model’s decisions.

A novel joint loss captures global explanation priors and combines cross-entropy and focal loss with uncertainty-based weighting, effectively balancing safety classification and explanation tagging.

The authors introduce a bias-countering synthetic explanation generation strategy to mitigate confirmation bias and supply token-level labels where human data are scarce.

Results cover both in-domain and out-of-domain settings on three datasets, reporting strong performance for prompt safety (safe/unsafe) and word-level explanation accuracy, supported by ablations of the joint loss.

**Weaknesses:**

The loss formulation in Equation 1 aggregates many terms without explicit balancing weights for most components, which risks scale-dominant terms steering optimization; adding learnable or tuned coefficients and reporting a sensitivity analysis would likely improve stability and performance.

According to Table 1, the Prompt baseline performs comparably to LEG and sometimes surpasses it, which weakens the claim that the carefully designed joint loss is the main source of improvement; statistical significance tests, effect sizes, and experiments isolating the loss’s contribution beyond the encoder are needed.

The ablation study in Table 6 yields dataset-dependent winners for different loss-term combinations, which suggests limited generalization of a single configuration; proposing a simplified default would strengthen confidence.

The latency comparison with Llama Guard 3 may be functionally uneven, because Llama Guard 3 (~57 ms) is already very fast and outputs fine-grained unsafe categories while LEG provides only a binary label; a fair comparison should match task granularity or extend LEG to multi-label predictions and evaluate end-to-end throughput.

**Questions:**

For the word-level label, is there a formal definition? A token that appears “harmful” in isolation may be non-harmful in other semantic contexts. Do you account for sentence-level semantics in the loss, and if so, how is the whole-sentence meaning incorporated?

Because the word labels are obtained from GPT-4o-mini, the supervision itself may not be fully faithful. After supervised learning, the faithfulness of the proposed method could therefore be upper-bounded by GPT-4o-mini. In addition, there is no baseline evaluation of GPT-4o-mini on the benchmarks, which makes the evaluation less comprehensive.

There exist explainable AI methods that produce reasons for a model decision, such as LRP (https://github.com/rachtibat/LRP-eXplains-Transformers). This approach yields token-level relevance scores with a single backpropagation pass over the guard model, covering all tokens—not only those deemed harmful—and providing both positive and negative contributions regardless of whether the final decision is safe or unsafe. This seems more faithful, whereas your method explains only when the decision is unsafe, outputs only “harmful” words, and does not indicate which tokens contribute more.

There is also a family of related methods like SHAP and its extensions that come with theoretical guarantees. How does your method compare against these baselines?

For the faithfulness evaluation in Section 5.3, is the metric standard? A more common approach is to mask words sequentially according to an importance metric and measure the resulting accuracy curve. In your case, you use classifier confidence as the metric; it would be helpful to compare the area under the accuracy curve (AUC) across methods.

---

> ### Author Response · Authors · 2025-12-03
> **Author’s Response to Reviewer GwUv (Part 1)**
>
> We thank the reviewer for taking the time to read our paper and provide their feedback. We address each point in detail below. Due to OpenReview’s character limits, our full response is divided into multiple comments. We respectfully request that the reviewer reconsider their score after reviewing our responses.
>
> > **Reviewer's comment:** “The loss formulation in Equation 1 aggregates many terms without explicit balancing weights for most components, which risks scale-dominant terms steering optimization; adding learnable or tuned coefficients and reporting a sensitivity analysis would likely improve stability and performance.”
>
> **Our Response:** We thank the reviewer for this comment. We would like to clarify that we explicitly mention in the paper that we use learnable parameters to balance the contribution of each task during training. Specifically, we adopt uncertainty-based weighting, which is a well-known technique in multitask learning. This is described in lines 287–292 of the paper. For the reviewer’s convenience, we include the relevant passage from our paper below:
>
> “Uncertainty-based weighting: In our joint loss function (equation 1) σ1 and σ2 are learnable uncertainty parameters that dynamically balance the contribution of each task during training. Effective loss weighting is essential in multi-task learning to balance the contribution of each task, preventing
> dominant or noisy tasks from skewing the optimization process. We adopt the uncertainty-based
> weighting method proposed by Kendall et al. (2018), where the loss of each task is scaled by a
> learnable parameter σ.”
>
>
> > **Reviewer's comment:** “According to Table 1, the Prompt baseline performs comparably to LEG and sometimes surpasses it, which weakens the claim that the carefully designed joint loss is the main source of improvement; statistical significance tests, effect sizes, and experiments isolating the loss’s contribution beyond the encoder are needed.”
>
> **Our Response:**
> We thank the reviewer for this comment. We would first like to clarify that we do not claim anywhere in the paper that the carefully designed joint loss is the main source of improvement. Instead, the performance of LEG arises from a combination of synthetic supervision, novel loss design, architecture, and multitask training.
> The custom joint loss function contributes positively to performance, as demonstrated in the ablation study in Appendix F and further discussed in our response to the next comment. However, it is not claimed to be the sole driver of the overall performance.
> Regarding the reviewer’s observation that the Prompt Baseline sometimes performs comparably to LEG, we provide a detailed comparison using the statistics from Table 1 below. When comparing models of equivalent size (LEG base vs. Prompt baseline base and LEG large vs. Prompt baseline large), LEG performs better in the majority of settings:
> In-domain:
>
>  * LEG-base: 0 out of 3 wins
>  * LEG-large: 3 out of 3 wins
>
>
> Out-of-domain:
>
> * LEG-base: 5 out of 6 wins
>  * LEG-large: 6 out of 6 wins
>
>
>
> These comparisons highlight that LEG consistently outperforms the Prompt Baseline in the majority of the cases, particularly in out-of-domain evaluations, which is essential for real-world robustness. Additionally, a key strength of LEG is that it provides faithful word-level explanations at a low computational cost, which the Prompt Baseline does not offer.
> Regarding statistical significance, all results reported in Tables 1 and 2  presented as the average performance across three independent runs, and we report the mean $\\pm$ standard deviation for each entry. This provides a stable and statistically reliable result.

---

> > ### Author Response · Authors · 2025-12-03
> > **Author’s Response to Reviewer GwUv (Part 2)**
> >
> > > **Reviewer’s comment:** The ablation study in Table 6 yields dataset-dependent winners for different loss-term combinations, which suggests limited generalization of a single configuration; proposing a simplified default would strengthen confidence.
> >
> > **Our Response:**
> > We thank the reviewer for this comment. Although we experimented with various configurations in the ablation study, our main novel contribution in the loss function lies in the introduction of $\delta_p$ and $\delta_w$​ (In the main paper, the term is correctly referred to as $\delta_t$​, but in Appendix F it was mistakenly written as $\delta_w$​. In the final version, we will correct this inconsistency and ensure that $\delta_t$ is used consistently throughout the paper). Since the model is trained as a multitask learning problem, the impact of these terms should not be evaluated in isolation. Instead, their effect should be assessed jointly, along with the associated focal loss components that modulate them.
> > To better illustrate the impact of our contribution, we summarize below two key configurations from Table 6:
> >
> > 1. Full loss, which includes both $\delta_p$ and $\delta_w$
> > 2. Ablated loss, where both $\delta_p$ and $\delta_w$​ (and their focal components) are removed.
> >
> >
> >
> > For each test set, we compare these two configurations across all three training datasets. The summarized results show that the full loss outperforms the ablated loss in 22 out of 36 cases, which represents a substantial performance advantage. This indicates that including $\delta_p$ and $\delta_w$​ leads to consistent improvements in the majority of the cases across datasets.
> > Therefore, the full loss serves as a reasonable default configuration and demonstrates the effectiveness of our contribution.
> >
> > \\[
> > \\scriptsize
> > \\renewcommand{\\arraystretch}{1.2}
> > \\begin{array}{|l|l|cccc|cccc|cccc|}
> > \\hline
> > \\text{Train set} & \\text{Ablation}
> > & \\rlap{~~~~~~~~~~~~~~~\\text{AEGIS2.0}} & & &
> > & \\rlap{~~~~~~~~~~\\text{WildGurdMix}} & & &
> > & \\rlap{~~~~~~~~~~\\text{Toxic-Chat0124}} & & & \\\\
> > & &
> > \\rlap{\\text{LEG-base}} & & \\rlap{\\text{LEG-large}} &
> > & \\rlap{\\text{LEG-base}} & & \\rlap{\\text{LEG-large}} &
> > & \\rlap{\\text{LEG-base}} & & \\rlap{\\text{LEG-large}} & \\\\
> > & &
> > PC & EC & PC & EC & PC & EC & PC & EC & PC & EC & PC & EC \\\\
> > \\hline
> > \\text{AEGIS2.0} & \\text{Full joint loss (all terms)}
> > & \\mathbf{86.5} & 76.38 & \\mathbf{87.75} & 79.71
> > & 74.76 & \\mathbf{60.84} & 78.68 & \\mathbf{67.46}
> > & 68.16 & \\mathbf{59.46} & \\mathbf{71.18} & \\mathbf{63.69} \\\\
> > & \\text{Remove $\\delta_p$, $\\delta_w$ and focal loss}
> > & 86.45 & 77.3 & 87.62 & 79.97
> > & 74.8 & 59.02 & 79.6 & 66.02
> > & 67.67 & 58.79 & 68.44 & 62.72 \\\\
> > \\hline
> > \\text{WildGuardMix} & \\text{Full joint loss (all terms)}
> > & \\mathbf{83.55} & \\mathbf{74.75} & \\mathbf{81.56} & 76.83
> > & \\mathbf{87.09} & 72.57 & \\mathbf{87.97} & 76
> > & 55.73 & 58.61 & \\mathbf{63.55} & \\mathbf{61.86} \\\\
> > & \\text{Remove $\\delta_p$, $\\delta_w$ and focal loss}
> > & 80.74 & 73.7 & 81.14 & 76.84
> > & 86.43 & 73.42 & 88.48 & 76.1
> > & 56.48 & 59.66 & 59.62 & 59.87 \\\\
> > \\hline
> > \\text{Toxic-Chat0124} & \\text{Full joint loss (all terms)}
> > & \\mathbf{78.99} & 48.15 & \\mathbf{76.38} & \\mathbf{51.99}
> > & \\mathbf{68.21} & \\mathbf{37.83} & \\mathbf{63.81} & 33.4
> > & 67.43 & \\mathbf{60.85} & \\mathbf{79.76} & \\mathbf{66.19} \\\\
> > & \\text{Remove $\\delta_p$, $\\delta_w$ and focal loss}
> > & 78.42 & 48.61 & 73.83 & 51.19
> > & 65.92 & 29.78 & 63.45 & 36.18
> > & 70.69 & 62.79 & 78.42 & 65.87 \\\\
> > \\hline
> > \\end{array}
> > \\]
> >
> > **Table:** Two key rows from Table 6 for each test set. In the table, PC indicates the performance of the prompt classifier, and EC indicates the performance of the explainability classifier.
> >
> >
> >
> > > **Reviewer’s comment:** “The latency comparison with Llama Guard 3 may be functionally uneven, because Llama Guard 3 (~57 ms) is already very fast and outputs fine-grained unsafe categories while LEG provides only a binary label; a fair comparison should match task granularity or extend LEG to multi-label predictions and evaluate end-to-end throughput.”
> >
> > **Our Response:**
> > While Llama Guard 3 outputs safety categories, our method provides built-in token-level explanations, which Llama Guard 3 does not support. LEG performs both safety classification and explanation in 7.81 ms (xs) to 14.57 ms (large), making it substantially lighter while offering more functionally complex outputs. Moreover, as shown in Table 1, LEG achieves higher detection performance than Llama Guard 3.
> > In practical deployments, a binary decision is sufficient to block unsafe prompts, and fine-grained categories rarely change downstream behavior. In contrast, explanations provide actionable value by enabling inspection, debugging, and monitoring of guardrail behavior. Thus, LEG offers stronger performance and richer functionality at significantly lower latency.

---

> > > ### Author Response · Authors · 2025-12-03
> > > **Author’s Response to Reviewer GwUv (Part 3)**
> > >
> > > > **Reviewer’s comment:**
> > > “Questions: For the word-level label, is there a formal definition? A token that appears “harmful” in isolation may be non-harmful in other semantic contexts. Do you account for sentence-level semantics in the loss, and if so, how is the whole-sentence meaning incorporated?”
> > >
> > > **Our Response:**
> > > Word labels are predicted in context, not in isolation. The explanation classifier operates on the final hidden states of a bidirectional transformer encoder (DeBERTa), where each token embedding reflects information from both its left and right context through self-attention. As a result, the model’s predictions for token-level unsafe labels are based on the full sentence semantics, not on the token alone. This ensures that words that are harmless in some contexts but harmful in others are classified appropriately based on their contextual meaning.
> > >
> > >
> > >
> > >
> > >
> > > > **Reviewer’s comment:**
> > > “Because the word labels are obtained from GPT-4o-mini, the supervision itself may not be fully faithful. After supervised learning, the faithfulness of the proposed method could therefore be upper-bounded by GPT-4o-mini. In addition, there is no baseline evaluation of GPT-4o-mini on the benchmarks, which makes the evaluation less comprehensive.”
> > >
> > > **Our Response:**
> > > In Appendix I, we added a human evaluation in which a human annotator labeled 50 randomly selected unsafe prompts from each of the three test sets by identifying the specific words that make the prompt unsafe. This resulted in a small but high-quality gold-labeled test set for token-level explanation evaluation.
> > >
> > > We evaluated both GPT-4o-mini and LEG on this human-labeled benchmark, and the results are shown in the table below. The results indicate that LEG-base and LEG-large outperform GPT-4o-mini on AEGIS-2.0 and WildGuardMix. On ToxicChat0124, while GPT-4o-mini performs slightly better, LEG-large is very close, with a difference of less than one percent.
> > >
> > > This demonstrates that, even though the explainability classifier is trained using silver labels derived from GPT-4o-mini, its performance is not upper-bounded by GPT-4o-mini, and LEG can learn explanation behavior that aligns more closely with human judgments. We will add this comparison to the Appendix.
> > >
> > > \\[
> > > \\scriptsize
> > > \\renewcommand{\\arraystretch}{1.2}
> > > \\begin{array}{|l|c|c|c|}
> > > \\hline
> > > & \\text{AEGIS 2.0} & \\text{WildGurdMix} & \\text{ToxicChat0124} \\\\
> > > \\hline
> > > \\text{GPT-4o-mini} & 53.28 & 54.87 & \\mathbf{50.96} \\\\
> > > \\text{LEG base} & 60.33 & \\mathbf{60.69} & 36.12 \\\\
> > > \\text{LEG large} & \\mathbf{66.11} & 58.39 & 50.36 \\\\
> > > \\hline
> > > \\end{array}
> > > \\]
> > > **Table:** Evaluation of the explainability classification on the human-annotated test set.
> > >
> > > > **Reviewer’s comment:** “There exist explainable AI methods that produce reasons for a model decision, such as LRP (https://github.com/rachtibat/LRP-eXplains-Transformers). This approach yields token-level relevance scores with a single backpropagation pass over the guard model, covering all tokens—not only those deemed harmful—and providing both positive and negative contributions regardless of whether the final decision is safe or unsafe. This seems more faithful, whereas your method explains only when the decision is unsafe, outputs only “harmful” words, and does not indicate which tokens contribute more.”
> > >
> > > **Our Response:**
> > > We appreciate the reviewer’s suggestion regarding post-hoc methods such as LRP. Our approach differs in that LEG learns explanations jointly with the unsafe prediction task, so the model is explicitly trained to identify the evidence that drives its own decisions. This joint optimization encourages causal alignment between the classifier and the explanation head, producing sparse, decision-focused rationales that directly correspond to the model’s safety prediction.
> > > By contrast, post-hoc attribution methods such as LRP approximate relevance after training and therefore do not influence how the model forms its decision boundary. As a result, relevance estimates may diverge from the features actually used during prediction, particularly in settings where harmful content is sparse or localized. In addition, LRP typically produces dense, real-valued scores for all tokens, which can be difficult to interpret, lack a clear threshold for action, and may be sensitive to architectural choices or numerical heuristics, limiting their suitability for safety workflows that require concise and actionable rationales.
> > >
> > > We did not include an empirical comparison because the current implementation of LRP does not support DeBERTa, and adapting it would require substantial time and engineering effort. If the paper is accepted, and if feasible within the camera-ready timeline, we will try to obtain results using LRP and compare them with our method.

---

> ### Author Response · Authors · 2025-12-03
> **Author’s Response to Reviewer GwUv (Part 4)**
>
> > **Reviewer’s comment:**
> There is also a family of related methods like SHAP and its extensions that come with theoretical guarantees. How does your method compare against these baselines?
>
> **Our Response:**
> We experimented with SHAP and report the results in the table below. We applied SHAP in the same setup as LIME with the Prompt baseline, where SHAP explains the predictions of the Prompt classifier. We will include this result in Table 2 of the paper as an additional baseline.
> Across all experimental settings, LEG substantially outperforms SHAP, often by large margins. This holds consistently for both in-domain and out-of-domain evaluation, and across all three datasets.
> These results are also consistent with prior findings that post-hoc explanation methods can struggle in settings where features are sparse, highly contextual, or adversarial, whereas models trained with integrated supervision can better align explanations with prediction behavior.
>
> \\[
> \\scriptsize
> \\renewcommand{\\arraystretch}{1.2}
> \\begin{array}{|l|l|c|c|c|}
> \\hline
> \\text{Train set} & \\text{Model} & \\text{AEGIS 2.0} & \\text{WildGurdMix} & \\text{ToxicChat0124} \\\\
> \\hline
>  & \\text{SHAP baseline base} & 41.07 & 25.73 & 9.03 \\\\
> \\text{AEGIS 2.0} & \\text{SHAP baseline large} & 45.94 & 32.16 & 20.944 \\\\
> & \\text{LEG base} & 76.95 & 60.40 & 59.78 \\\\
>  & \\text{LEG large} & \\mathbf{79.60} & \\mathbf{66.66} & \\mathbf{63.18} \\\\
> \\hline
>  & \\text{SHAP baseline base} & 37.21 & 33.44 & 7.28 \\\\
> \\text{WildGurdMix} & \\text{SHAP baseline large} & 43.99 & 39.88 & 15.76 \\\\
>  & \\text{LEG base} & 74.28 & 73.16 & 58.86 \\\\
>  & \\text{LEG large} & \\mathbf{76.93} & \\mathbf{75.83} & \\mathbf{61.56} \\\\
> \\hline
>  & \\text{SHAP baseline base} & 33.12 & 25.46 & 11.78 \\\\
> \\text{ToxicChat0124} & \\text{SHAP baseline large} & 40.88 & 30.56 & 21.47 \\\\
>  & \\text{LEG base} & 45.91 & 33.77 & 60.66 \\\\
>  & \\text{LEG large} & \\mathbf{52.77} & \\mathbf{38.07} & \\mathbf{65.99} \\\\
> \\hline
> \\end{array}
> \\]
> **Table:** Comparison of the performance of the explainability classification of SHAP and LEG.
>
>
> > **Reviewer’s comment:** For the faithfulness evaluation in Section 5.3, is the metric standard? A more common approach is to mask words sequentially according to an importance metric and measure the resulting accuracy curve. In your case, you use classifier confidence as the metric; it would be helpful to compare the area under the accuracy curve (AUC) across methods.
>
> **Our Response: **
>
> The faithfulness evaluation in Section 5.3 follows a widely used perturbation-based methodology in NLP. Specifically, we remove the top-k tokens identified as most important by the explanation classifier and measure the resulting degradation in the prompt classifier’s performance. This approach directly tests whether the tokens identified as unsafe by the explainer are causally necessary for the prompt classifier’s decisions. Such deletion-based tests are standard in the rationale and explanation literature, which evaluates explanations by removing tokens and measuring how prediction quality changes under perturbation.
>
> This evaluation method was formalized and standardized for NLP by the ERASER benchmark (https://aclanthology.org/2020.acl-main.408.pdf), and has been widely adopted in many subsequent papers.
> In our setup, token importance is computed using the model’s predicted probabilities, which is a common and principled choice for ranking tokens (also used in ERASER and related perturbation-based attribution work). Our results show a consistent performance decline as more top-ranked unsafe tokens are masked, demonstrating both causal sensitivity and monotonicity, two key indicators of explanation faithfulness.
>
> A commonly used scalar summary of perturbation-based evaluation is Area Over the Perturbation Curve (AOPC), which computes the mean drop in performance across deletions. This is particularly useful when comparing the faithfulness of multiple explanation methods or baselines, because it summarizes the perturbation curves into a single quantitative score. In our setting, however, the goal is not to compare our method against alternative explanation techniques, but rather to assess whether the explanation tokens produced by our method are causally important to the prompt classifier head. As shown in Table 3, masking the top-1, top-2, and top-3 explanation tokens produces substantial and progressively larger performance drops, indicating that the model relies heavily on these tokens. This monotonic and large degradation directly demonstrates faithfulness.

---

### Official Review · Reviewer_SUWd · 2025-10-27

**Soundness:** 3
**Presentation:** 3
**Contribution:** 3
**Rating:** 6
**Confidence:** 4

**Summary:**

This paper introduces a new external, lightweight guardrail for prompt safety. It outperforms existing guardrails on prompt classification and can highlight components of the prompt that are potentially unsafe. Notably, it's much smaller than current LLM-based guardrails such as LlamaGuard and WildGuard.

**Strengths:**

The model design is original and novel. The quality of evaluation is relatively high, with reasonable baselines and benchmarks. Table 1 highlights strong prompt classification performance compared to existing guardrails, demonstrating the effectiveness of the proposed model. The writing is clear.

**Weaknesses:**

Overall I like the idea and enjoyed reading the paper. However, I see the following two weaknesses:

1. Since the loss function is impacted by the overall frequencies of tokens in safe vs unsafe contexts in the training set, the trained model might be sensitive to these keywords. For example, a prompt might contain typically unsafe keywords while being safe overall. Evaluation on such cases will provide more information about the trustworthiness of LEG. OR-Bench (https://arxiv.org/pdf/2405.20947?) might be a good resource for this.

2. Even though the explainability classification is human-interpretable, the mechanistic relationship between the identified keywords and the prompt classification result is not directly interpretable, despite their correlations shown in Table 3. Thus, LEG's prompt classification decisions aren't fully explainable. Since it's not feasible to address this within the conference timeline, please acknowledge it as a limitation beyond stating that "the generated explanations are faithful".

**Questions:**

1. An informative baseline to consider adding is a frontier/SOTA LM for zero-shot prompt classification and explainability classification - how does LEG compare to it?

2. Even though the three prompt datasets are not highly similar lexically, they might share topical similarity. For example, they might use instances that fall under a similar set of risk categories. Does LEG maintain robust generalization performance between risk categories, rather than between datasets?

3. Lines 197-199: how often did the LLM fail the confirmation bias test?

4. Is the base model for LEG pre-trained? Or is the model only trained on prompt safety datasets?

---

> ### Author Response · Authors · 2025-12-04
> **Author’s Response to Reviewer SUWd (Part 1)**
>
> We thank the reviewer for taking the time to read our paper and provide their feedback. We address each point in detail below. Due to OpenReview’s character limits, our full response is divided into multiple comments. We respectfully request that the reviewer reconsider their score after reviewing our responses.
>
> > **Reviewer's comment:** “1.Since the loss function is impacted by the overall frequencies of tokens in safe vs unsafe contexts in the training set, the trained model might be sensitive to these keywords. For example, a prompt might contain typically unsafe keywords while being safe overall. Evaluation on such cases will provide more information about the trustworthiness of LEG. OR-Bench (https://arxiv.org/pdf/2405.20947?) might be a good resource for this.”
>
>
> **Our Response:**
> LEG operates on the final hidden states of a bidirectional transformer encoder (DeBERTa), where each token embedding incorporates information from both its left and right context through self-attention. Consequently, token-level predictions are based on sentence-level semantics, not on surface forms alone, which allows the model to treat the same word differently depending on its contextual meaning.
> The loss function includes two additional terms, $\\delta_t$​ and $\\delta_p$​, which introduce a penalty based on the polarity of a word toward the safe or unsafe class, computed from overall frequency statistics in the training data. This polarity signal modulates the magnitude of the loss but does not determine the label. The classifier is always trained on contextualized embeddings paired with ground-truth labels. So supervision remains contextual even when the penalty is applied. In other words, polarity affects how strongly the model is encouraged to correct a mistake, but the model does not learn to classify tokens based on isolated frequency information.
> We acknowledge that evaluating prompts where typically “unsafe” keywords appear in benign contexts is valuable for assessing robustness. Elements of this scenario are already present in datasets such as ToxicChat0124, which contain adversarial, misleading, and benign uses of unsafe terminology, and LEG maintains strong performance under such conditions (Table 1).
>
> > **Reviewer's comment:** “2.Even though the explainability classification is human-interpretable, the mechanistic relationship between the identified keywords and the prompt classification result is not directly interpretable, despite their correlations shown in Table 3. Thus, LEG's prompt classification decisions aren't fully explainable. Since it's not feasible to address this within the conference timeline, please acknowledge it as a limitation beyond stating that "the generated explanations are faithful".
>
> **Our Response:**
> We appreciate the reviewer’s point about the distinction between explanation faithfulness and mechanistic interpretability. Our work focuses on generating token-level rationales that are causally aligned with model predictions, rather than uncovering the internal mechanisms that produce those predictions. As shown in Table 3, perturbation-based testing demonstrates that removing top-ranked tokens leads to a consistent performance drop.
> This is a widely accepted method for evaluating faithfulness in explainable AI. It provides empirical evidence that the identified tokens are causally relevant to the model’s output.
> However, faithfulness is different from mechanistic interpretability. Mechanistic interpretability seeks to understand how internal components of the model contribute to its predictions. Mechanistic interpretability requires systematically reverse-engineering internal computations, which is beyond the scope of our current work. We will clarify this distinction in the paper and outline mechanistic interpretability as a promising direction for future research.

---

> > ### Author Response · Authors · 2025-12-04
> > **Author’s Response to Reviewer SUWd (Part 2)**
> >
> > > **Reviewer's comment:** “An informative baseline to consider adding is a frontier/SOTA LM for zero-shot prompt classification and explainability classification - how does LEG compare to it?”
> >
> > **Our Response:**
> > We appreciate the reviewer’s suggestion to include zero-shot evaluations of large language models. We experimented with zero-shot prompting using GPT-4o, GPT-4o-mini, and the open-source Llama-3.1-8B-instruct model, and we present the results in the table below.
> >
> > \\[
> > \\begin{array}{lccc}
> > \\hline
> > \\text{Models} & \\textbf{AEGIS 2.0} & \\textbf{WildGuardMix} & \\textbf{ToxicChat0124} \\\\
> > \\hline
> > \\text{GPT-4o-mini} & 83.94 & 82.92 & 63.95 \\\\
> > \\text{GPT-4o} & 84.92 & 85.87 & 69.79 \\\\
> > \\text{Llama-3.1 8B instruct} & 79.23 & 76.52 & 53.20 \\\\
> > \\text{LEG (Best in-domain performance)} & \\mathbf{87.54} & \\mathbf{87.74} & \\mathbf{78.58} \\\\
> > \\text{LEG (Best out-of-domain performance)} & 82.07 & 79.04 & 69.98 \\\\
> > \\hline
> > \\end{array}
> > \\]
> > **Table:** Comparison of LEG model with zero-shot LLM baselines.
> >
> >
> >
> > The comparison with GPT variants shows that:
> >
> > * Our in-domain performance is better than both GPT-4o and GPT-4o-mini across all three datasets (AEGIS 2.0, WildGuardMix, and ToxicChat0124).
> >
> >
> > * Our out-of-domain performance is also comparable or better. On AEGIS 2.0 and WildGuardMix, our model is close to the GPT results, while on ToxicChat0124 our LEG model outperforms both GPT variants. This is particularly significant because ToxicChat contains more challenging, noisy, and jailbreak-style prompts. These results show that our proposed method outperforms both GPT-4o and GPT-4o-mini in out-of-domain evaluation under these challenging conditions.
> >
> > However, it is important to emphasize that GPT models are not realistic choices for guardrail applications. First, a guardrail must be lighter than or comparable in size to the model it is protecting. Using GPT models as a safety filter for an LLM would result in a pipeline where the safety checker is significantly larger and more costly than the model being safeguarded. This is not practical from the standpoints of latency, system design, or compute resources. Second, GPT models are closed-source and accessible only through a paid API, which makes per-prompt safety checks financially unrealistic for any real deployment scenario. In contrast, effective guardrails must be inexpensive, efficient, and easily deployable in local or offline environments.
> >
> > An open-source small model, such as Llama-3.1-8B-Instruct could be a realistic choice for a guardrail model. However, the results in the above Table show that our method consistently outperforms Llama-3.1-8B-instruct when LEG is trained in both in-domain and out-of-domain settings, while remaining lightweight and providing faithful explanations. We will add this discussion to the appendix in the final version of the paper.

---

> > > ### Author Response · Authors · 2025-12-04
> > > **Author’s Response to Reviewer SUWd (Part 3)**
> > >
> > > > **Reviewer's comment:** “Even though the three prompt datasets are not highly similar lexically, they might share topical similarity. For example, they might use instances that fall under a similar set of risk categories. Does LEG maintain robust generalization performance between risk categories, rather than between datasets?”
> > >
> > > **Our Response:**
> > > To evaluate whether LEG can generalize across risk categories rather than only across datasets, we conducted an additional experiment on the WildGuardMix dataset. In this dataset, each unsafe prompt is annotated with one of 14 risk categories, allowing us to study cross-category generalization.
> > > We trained LEG base under two settings:
> > >
> > > 1. Full training: trained on the complete WildGuardMix training set.
> > > 2. Category-excluded training: trained after removing all instances of four randomly selected unsafe categories (shown in red in the table). These four categories were therefore completely unseen during training.
> > >
> > >
> > > The objective of this experiment is to test whether LEG can correctly classify prompts from risk categories not observed during training, rather than relying on lexical overlap.
> > > The table below shows test-set performance under both settings. For the four excluded categories, LEG achieves performance that is very close to the full-data setting in three out of four excluded categories. The only notable drop occurs for “copyright violations”, where prompt classification performance decreases to 62.2%, which is still reasonable given that this category was entirely unseen during training. For the other three categories, performance remains similar across both settings, indicating that LEG can generalize to unseen risk categories.
> > >
> > > Across both training settings, LEG maintains high performance across nearly all categories, demonstrating that the model does not simply memorize category-specific lexical cues. Instead, LEG appears to learn higher-level semantic features that enable transfer to unseen types of unsafe categories.
> > > Overall, this experiment indicates that LEG maintains robust generalization between risk categories, not just between datasets. Even when entire categories are excluded during training, LEG continues to perform well on those categories at test time. If accepted, we will include this result in the Appendix.
> > >
> > > \\[
> > > \\scriptsize
> > > \\renewcommand{\\arraystretch}{1.25}
> > > \\begin{array}{|l|cc|cc|}
> > > \\hline
> > > \\text{Unsafe Category}
> > > & \\rlap{~~~~~\\text{Setting 1}} & &
> > >   \\rlap{~~\\text{Setting 2}} & \\\\
> > > & \\text{Prompt} & \\text{Explainability} & \\text{Prompt} & \\text{Explainability} \\\\
> > > \\hline
> > > \\text{causing material harm by disseminating misinformation} & 97.7 & 70.27 & 98.9 & 70.3 \\\\
> > > \\textcolor{red}{\\text{copyright violations}} & 87.3 & 47.7 & 62.2 & 40.4 \\\\
> > > \\text{cyberattack} & 98.9 & 70.3 & 98.9 & 76.4 \\\\
> > > \\text{defamation encouraging unethical or unsafe actions} & 98.9 & 69.9 & 100 & 74.0 \\\\
> > > \\textcolor{red}{\\text{disseminating false or misleading information, encouraging disinformation campaigns}} & 98.9 & 69.9 & 98.9 & 67.3 \\\\
> > > \\textcolor{red}{\\text{fraud assisting illegal activities}} & 89.9 & 76.8 & 88.9 & 74.1 \\\\
> > > \\text{mental health over reliance crisis} & 94.4 & 70.4 & 94.4 & 68.6 \\\\
> > > \\text{others} & 79.8 & 85.8 & 82.6 & 86.1 \\\\
> > > \\text{private information individual} & 87.5 & 75.8 & 87.5 & 74.1 \\\\
> > > \\text{sensitive information organization government} & 92.5 & 75.0 & 92.5 & 77.3 \\\\
> > > \\text{sexual content} & 88.1 & 76.0 & 88.1 & 74.9 \\\\
> > > \\text{social stereotypes and unfair discrimination} & 77.4 & 73.0 & 81.2 & 75.7 \\\\
> > > \\text{toxic language hate speech} & 100 & 78.5 & 100 & 76.6 \\\\
> > > \\textcolor{red}{\\text{violence and physical harm}} & 98.7 & 67.6 & 100 & 67.7 \\\\
> > > \\hline
> > > \\end{array}
> > > \\]
> > >
> > > **Table:**  Performance of LEG-base on the WildGuardMix test set when trained on the full training set (Setting 1) versus training with four categories excluded (Setting 2). The four red-marked categories were excluded from the training data in Setting 2. Reported values are F1 scores for both prompt classification and explanation classification.
> > >
> > >
> > > > **Reviewer's comment:** “Lines 197-199: how often did the LLM fail the confirmation bias test?”
> > >
> > > **Our Response:** Using GPT-4o-mini as the LLM, the model failed the confirmation bias test in 34.3% of the instances in AEGIS2.0, 33.3% in WildGuardMix, and 14.2% in ToxicChat0124.
> > >
> > > > **Reviewer's comment:** “Is the base model for LEG pre-trained? Or is the model only trained on prompt safety datasets?”
> > >
> > > **Our Response:** The base model for LEG is a pre-trained encoder (DeBERTa). We fine-tune it, along with the additional components of our architecture, using prompt safety training datasets. This follows standard NLP practice, where a pre-trained language model is adapted to a domain-specific task through fine-tuning.

---

### Official Review · Reviewer_TLxi · 2025-11-01

**Soundness:** 2
**Presentation:** 2
**Contribution:** 2
**Rating:** 2
**Confidence:** 4

**Summary:**

The paper proposes a lightweight explainable guardrail (LEG) method for the classification of unsafe prompts. LEG uses a multi-task learning architecture to jointly learn a prompt classifier and an explanation classifier, where the latter labels prompt words that explain the safe/unsafe overall decision. It demonstrates competitive performance with a small number of parameters.

**Strengths:**

- The loss design is reasonable and the ablation study verifies the effectiveness of each component.
- The small training and inference cost, due to the small base model.

**Weaknesses:**

- Strong baselines are missing. For example, the prompt guard model trained by meta, which has strong detection performance.
- Limited contribution. The prompt guard model trained by meta also has a small number of parameters (86M), so the contribution of this paper is limited.
- The evaluation of using prompted large language models is missing. For example, using GPT-4o may achieve a strong performance.

**Questions:**

Why do you choose DeBERTa as the base model instead of causal language model?

---

> ### Author Response · Authors · 2025-11-27
> **Author’s Response to Reviewer TLxi (Part 1)**
>
> We thank the reviewer for taking the time to read our paper and provide their feedback. We address each point in detail below. Due to OpenReview’s character limits, our full response is divided into multiple comments. We respectfully request that the reviewer reconsider their score after reviewing our responses.
>
> > **Reviewer's comment:** “Strong baselines are missing. For example, the prompt guard model trained by meta, which has strong detection performance.”
>
> **Our Response:** We thank the reviewer for this comment. We would like to clarify that our paper does include a direct comparison with Llama Prompt Guard 2, the most recent version of the prompt guard model released by Meta.
> The reviewer can see the results in Table 1 (rows 5–6). Our paper reports model size as backbone parameters + embedding parameters for all models. Meta’s documentation for Llama Prompt Guard 2 reports only the backbone parameter size. To avoid inconsistency across baselines, we reported  backbone parameters + embedding parameters for Llama Prompt Guard 2 as well.
>
> This leads to the following alignment:
> * Llama Prompt Guard 2 22M (22M backbone + 48M embeddings = 70M total) &rarr; reported as 70M in Table 1.
> * Llama Prompt Guard 2 86M (86M backbone + 190M embeddings = 276M total) &rarr; reported as 276M in Table 1.
>
>
> So, the Llama Prompt Guard 2 22M is directly comparable to our LEG xs (22M backbone + 48M embeddings), and Llama Prompt Guard 2 86M is directly comparable to our LEG base (86M backbone + 98M embeddings).
>
> As shown in Table 1 of the paper, both versions of Llama Prompt Guard 2 underperform significantly compared to all LEG variants.
> The best performance of Llama Prompt Guard 2 (86M, reported as 276M) on the AEGIS2.0 dataset is only 8.5%, whereas our much smaller LEG xs model achieves an out-of-domain F1 score of 81.64%. We observe similarly large performance gaps on both the WildGuardMix and Toxic-Chat0124 datasets, further demonstrating that our method substantially outperforms Prompt Guard 2 despite using comparable or smaller model sizes. In addition to the strong performance, our method provides built-in explanations, a capability that Llama Prompt Guard models do not offer.
>
> We hope this clarifies that we did compare our method against the Prompt Guard models in Table 1, and that both our results and the integrated explainability feature make LEG clearly superior to the Prompt Guard models.
>
>
> > **Reviewer's comment:** “Limited contribution. The prompt guard model trained by meta also has a small number of parameters (86M), so the contribution of this paper is limited.”
>
> **Our Response:** As discussed in our response to the reviewer’s first comment, our empirical results show that both versions of Llama Prompt Guard 2 perform very poorly across all three datasets, even though they have similar or larger parameter counts compared to our smallest model. In addition, Prompt Guard 2 does not provide built-in explanations, which are essential in many safety-critical or regulated applications.
>
> In contrast, the contributions of our work extend well beyond model size. Specifically:
> 1. Faithful built-in explanations, validated through a perturbation-based faithfulness test (Table 3). Current guardrail models, including Prompt Guard 2, do not provide such explanations.
> 2. A novel synthetic data generation framework that capitalizes on confirmation bias to produce reliable word-level explanations. This approach directly addresses the lack of supervised explainability data in the guardrail domain and offers a novel path for generating high-quality synthetic training data. It also makes the overall training process significantly less expensive compared to relying on human annotators.
> 3. A multi-task learning framework with a novel joint loss function.
> 4. A lightweight architecture that achieves high performance with lower latency and fewer parameters than existing guardrail systems.
>
> To the best of our knowledge, no existing guardrail model combines lightweight architecture, strong cross-dataset performance, and faithful explanations achieved through multi-task learning and synthetic explanation supervision within a single framework. For this reason, the contribution of our work is substantial and fills an important gap in the current guardrail literature.
>
> We kindly ask the reviewer to consider our contributions in light of these points.

---

> ### Author Response · Authors · 2025-11-27
> **Author’s Response to Reviewer TLxi (Part 2)**
>
> > **Reviewer's comment:** “The evaluation of using prompted large language models is missing. For example, using GPT-4o may achieve a strong performance.”
>
> **Our Response:** We appreciate the reviewer’s suggestion to include prompted evaluations of large language models. We experimented with zero-shot prompting using GPT-4o, GPT-4o-mini, and the open-source Llama-3.1-8B instruct model, and we present the results in the table below.
>
> \\[
> \\begin{array}{lccc}
> \\hline
> \\text{Models} & \\textbf{AEGIS 2.0} & \\textbf{WildGuardMix} & \\textbf{ToxicChat0124} \\\\
> \\hline
> \\text{OpenAI Moderation API} & 37.8 & 12.1 & 61.41 \\\\
> \\text{GPT-4o-mini} & 83.94 & 82.92 & 63.95 \\\\
> \\text{GPT-4o} & 84.92 & 85.87 & 69.79 \\\\
> \\text{Llama-3.1 8B instruct} & 79.23 & 76.52 & 53.20 \\\\
> \\text{LEG (Best in-domain performance)} & \\mathbf{87.54} & \\mathbf{87.74} & \\mathbf{78.58} \\\\
> \\text{LEG (Best out-of-domain performance)} & 82.07 & 79.04 & 69.98 \\\\
> \\hline
> \\end{array}
> \\]
> **Table:** Comparison of LEG model with zero-shot LLM baselines.
>
>
>
> The comparison shows that:
>
> * Our in-domain performance is better than both GPT-4o and GPT-4o-mini across all three datasets (AEGIS 2.0, WildGuardMix, and ToxicChat0124).
>
>
> * Our out-of-domain performance is also comparable or better. On AEGIS 2.0 and WildGuardMix, our model is close to the GPT results, while on ToxicChat0124 our LEG model outperforms both GPT variants. This is particularly significant because ToxicChat contains more challenging, noisy, and jailbreak-style prompts. These results show that our proposed method outperforms both GPT-4o and GPT-4o-mini in out-of-domain evaluation under these challenging conditions.
>
>
> However, it is important to emphasize that GPT models are not realistic choices for guardrail applications.
>
> First, a guardrail must be lighter than or comparable in size to the model it is protecting. Using GPT models as a safety filter for an LLM would result in a pipeline where the safety checker is significantly larger and more costly than the model being safeguarded. This is not practical from the standpoints of latency, system design, or compute resources. Second, GPT models are closed-source and accessible only through a paid API, which makes per-prompt safety checks financially unrealistic for any real deployment scenario. In contrast, effective guardrails must be inexpensive, efficient, and easily deployable in local or offline environments.
>
>
> In the table above, we also include two additional models that can realistically be used as guardrails for large language models. The first is an open-source small model, Llama-3.1-8B-instruct. The results show that our method consistently outperforms Llama-3.1-8B-instruct when LEG is trained in both in-domain and out-of-domain settings. The second is the OpenAI Moderation API, which is OpenAI’s official guardrail classifier and a practical choice for prompt safety checking. This model is a text classifier specifically designed for safety monitoring. However, as shown in the table, its performance is substantially lower than that of our models across all datasets.
>
> Taken together, these results demonstrate two important points. First, although large models such as GPT-4o can perform reasonably well when prompted, they are not realistic or cost-effective guardrail solutions because they are closed source, expensive to query, and in most cases far larger than the models they would be protecting. Second, our paper compares LEG with eight recent and realistic guardrail models, and LEG achieves the strongest overall performance in both in-domain and out-of-domain settings while also providing faithful explanations.
>
> Although we did not include GPT-4o as a baseline in the main paper due to its impracticality for this use case, we included all major recent baselines in the guardrail domains and showed that LEG performs strongly in comparison to existing guardrail systems. The additional results presented here further show that our method is also comparable to strong NLP baselines such as GPT-4o and GPT-4o-mini, even though these models are not realistic choices for guardrail applications.
> We will add this discussion and the comparison of GPT-4o and GPT-4o-mini with our LEG model to the appendix of the paper.

---

> > ### Author Response · Authors · 2025-11-27
> > **Author’s Response to Reviewer TLxi (Part 3)**
> >
> > > **Reviewer's comment:** “Questions: Why do you choose DeBERTa as the base model instead of a causal language model?”
> >
> > **Our Response:** The main goal of this paper is to design a guardrail model that is lightweight and efficient. A guardrail is intended to protect a causal language model from harmful or unsafe use cases, and therefore, it should introduce only a small computational overhead on top of the already expensive LLM text generation process. Using a causal large language model as a guardrail would require deploying one large model to monitor another, which is not a viable design choice in terms of latency, compute cost, and memory requirements.
> >
> > We therefore adopt a small encoder-based approach to keep the guardrail lightweight and low-latency. DeBERTa is a strong choice among existing encoders because its disentangled attention and enhanced positional embeddings provide strong language understanding while keeping the model compact and computationally efficient. This makes it well-suited for real-time prompt safety checking. As shown in Table 1 of the paper, our lightweight encoder-based models perform as well as or better than expensive LLM-based guardrails such as LlamaGuard, AegisGuard, and WildGuard. These results demonstrate that a small encoder-only architecture can match or exceed the performance of much larger causal models while offering significantly lower inference cost.
> >
> >
> > We appreciate the reviewer’s time and consideration, and in light of this discussion, we respectfully request that the reviewer reconsider the rating/score.

---

### Note · Authors · 2026-01-02

I have read and agree with the venue's withdrawal policy on behalf of myself and my co-authors.